# Learning by Self-Explaining

**Wolfgang Stammer**[1,2]                    *wolfgang.stammer@cs.tu-darmstadt.de*

**Felix Friedrich**[1,2]                    *friedrich@cs.tu-darmstadt.de*

**David Steinmann**[1,2]                    *david.steinmann@tu-darmstadt.de*

**Manuel Brack**[1,2,4]                    *brack@cs.tu-darmstadt.de*

**Hikaru Shindo**[1]                    *hikaru.shindo@cs.tu-darmstadt.de*

**Kristian Kersting**[1,2,3,4]                    *kersting@cs.tu-darmstadt.de*

[1]Artificial Intelligence and Machine Learning Group, TU Darmstadt
[2]Hessian Center for Artificial Intelligence (hessian.AI), Darmstadt
[3]Centre for Cognitive Science, TU Darmstadt
[4]German Center for Artificial Intelligence (DFKI)

**Reviewed on OpenReview:** *https://openreview.net/forum?id=bpjU7rLjJ7*

## Abstract

Much of explainable AI research treats explanations as a means for model inspection. Yet, this neglects findings from human psychology that describe the benefit of self-explanations in an agent's learning process. Motivated by this, we introduce a novel workflow in the context of image classification, termed Learning by Self-Explaining (LSX). LSX utilizes aspects of self-refining AI and human-guided explanatory machine learning. The underlying idea is that a learner model, in addition to optimizing for the original predictive task, is further optimized based on explanatory feedback from an internal critic model. Intuitively, a learner's explanations are considered "useful" if the internal critic can perform the same task given these explanations. We provide an overview of important components of LSX and, based on this, perform extensive experimental evaluations via three different example instantiations. Our results indicate improvements via Learning by Self-Explaining on several levels: in terms of model generalization, reducing the influence of confounding factors, and providing more task-relevant and faithful model explanations. Overall, our work provides evidence for the potential of self-explaining within the learning phase of an AI model.

## 1 Introduction

Self-reflection is considered an important building block of human intelligence and a crucial component in the learning process of humans (Gläser-Zikuda, 2012; Ellis et al., 2014). In fact, one aspect of self-reflection—self-explaining—has been identified in several psychological studies as greatly beneficial for the overall learning, problem-solving and comprehension abilities of human subjects (Chi, 2018; Chi et al., 1981; 1994; Chamberland & Mamede, 2015; Belobrovy, 2018; Larsen et al., 2013; Kwon & Jonassen, 2011; Bisra et al., 2018). Accordingly, self-explanations have been suggested to act as a means of making initially implicit knowledge explicit and therefore to represent an important component for critical *self-refinement*.

Indeed, recent works in machine learning (ML) research have picked up on the idea of self-refining. While some are directly inspired by findings from human studies (Madaan et al., 2023), others utilize the potential of pre-trained large language models (LLMs), *e.g.*, on the topics of self-debiasing (Schick et al., 2021), self-instructing (Wang et al., 2023) and self-rewarding (Yuan et al., 2024). Although these works are quite specific

Figure 1: (left) Current (self-)refinement machine learning utilizes (I.) forms of *self-supervised* model refinement (*e.g.*, self-rewarding). On the other hand, it also relies on (II.) explanatory interactive learning (XIL) which utilizes human feedback via *explanations* for model refinement. (right) In contrast, we introduce **Learning by Self-Explaining** which integrates ideas from both research fields into one approach as *explanation*-based *self*-refinement (I + II). A model in LSX consists of two submodels, a *learner* and (internal) *critic*, and performs refinement via four modules (FIT, EXPLAIN, REFLECT, REVISE, *cf.* Alg. 1). The learner is optimized for a base task in FIT (*e.g.*, image classification), after which it provides explanations to its decisions in EXPLAIN. In the REFLECT module, the critic assesses how useful the explanations are for performing the base task. The resulting feedback from the critic is used to REVISE the learner.

in their form of self-refinement (*cf.* survey by Pan et al. (2023)) and far from the general idea of self-reflection from human psychology, they do provide valuable first steps for more *reflective* AI models. However, none of these focus on the value and potential of explanations as the basis and means of such reflective processes.

On the other hand, research on interactive machine learning (Teso et al., 2023; Gao et al., 2024), such as explanatory interactive learning (XIL) (Teso & Kersting, 2019; Schramowski et al., 2020), has long identified the value of explanations as a means of communication between human users and AI models, particularly, as a very successful means for model refinement. However, hereby explanations are only leveraged for refinement through human guidance.

In this work, we introduce Learning by Self-Explaining (LSX), a novel workflow that combines the ideas of self-refining ML and XIL (*cf.* Fig. 1 (left)) by leveraging explanations in the learning process of a model prior to any form of human explanatory feedback. In LSX (*cf.* Fig. 1 (right)) a model consists of two submodels, a *learner* and an internal *critic*, and undergoes four distinct learning modules (FIT, EXPLAIN, REFLECT, REVISE). In FIT, the learner is trained on a base task (*e.g.*, image classification) upon which it provides explanations for its predictions (EXPLAIN). The critic next assesses the quality of these explanations for performing the base task (REFLECT). Intuitively, a "useful" explanation should thereby provide important information for the task at hand. Finally, the critic's feedback is used to revise the learner (REVISE) and the EXPLAIN, REFLECT and REVISE loop is repeated, if needed. Overall, we consider the two submodels to constitute a collective model, whereby both work jointly to improve on the same base task. However, at inference time, only the learner provides the final task predictions (*cf.* Fig. 1 (right)).

We introduce LSX in the context of image classification and provide experimental evaluations via multiple datasets and evaluation metrics to investigate the benefits of learning by self-explaining. Specifically, these evalautions focus on the generalization performances of a learner, mitigating confounding behavior, consolidating explanations and explanation faithfulness. We perform these evaluations on three diverse example instantiations where the learner is based on a convolutional neural network (CNN), a neuro-symbolic (NeSy), and a vision-language model (VLM), respectively.

Overall, our contributions are the following: (i) We introduce LSX, a novel learning workflow in the spirit of XIL and self-refining ML in which a model refines itself by assessing its explanations. (ii) In the context of image classification, we introduce several example instantiations for LSX thereby illustrating different

---

**Algorithm 1** Learning by Self-Explaining in pseudocode. Given: the two submodels, a learner model, $f$, and internal critic model, $c$, dataset $\bar{X} = (X, Y)$ (*i.e.*, input images and corresponding class labels), $\bar{X}_c \subseteq \bar{X}$, iteration budget $T$ and base task, *i.e.*, image classification.

---

$f \leftarrow \text{FIT}(f, \bar{X})$ {Learner optimized for base task}
**repeat**
    `explanations` $\leftarrow \text{EXPLAIN}(f, \bar{X}_c)$ {Obtain explanations from learner for examples $\bar{X}_c$}
    `feedback` $\leftarrow \text{REFLECT}(c, \bar{X}_c, \texttt{explanations})$ {Critic provides feedback on quality of `explanations`}
    $f \leftarrow \text{REVISE}(f, \bar{X}, \texttt{feedback})$ {Learner is updated given feedback from critic}
**until** budget $T$ is exhausted or $f$ converged
**return** $f$

---

model and learning configurations. (iii) We provide extensive experimental evidence on various datasets and evaluation metrics, illustrating the potential of LSX for model refinement.

We proceed as follows. In section 2, we formally introduce LSX. For our experimental evaluations in section 3, we introduce several example instantiations for LSX and provide results for multiple datasets, metrics, and ablations. We wrap up our work with an extensive discussion of our results as well as related work and leave the reader with a final conclusion.

## 2 Learning by Self-Explaining (LSX)

In the following, we introduce Learning by Self-Explaining in the context of image classification and present an overview of its important components and modules. Let us first provide some background notations.

In LSX, a model is comprised of two submodels: the *learner* submodel, denoted as $f$, and the *internal critic*, denoted as $c$ (*cf.* Fig. 1 (right)). Further, let $x \in X$ be an image, with $X := \{x_i\}_{i=1}^N$, and with corresponding class label $y \in Y$ with $Y \subseteq \mathbb{N}_{\leq K}^N$ for $K$ classes. Hereby, $N \in \mathbb{N}$ denotes the number of samples. We denote $\bar{X} := (X, Y)$ as the joint set of images and corresponding labels and $\bar{X}_c \subseteq \bar{X}$ as a subset of this that is specifically provided to the critic (details below).

While the learner performs an underlying base task, *e.g.*, image classification, the critic's role is to provide feedback on the learner's explanations within the task. Overall, the learner represents a trainable predictive model (*e.g.*, a CNN). The critic also represents a predictive model which, however, must not necessarily be learnable. Its specific model type should be chosen depending on constraints such as the type of explanation that the learner provides (will be discussed below). At inference, the learner provides the overall model's final predictions (*cf.* Fig. 1 (right)).

The general procedure of LSX can be described via four modules (inspired by Friedrich et al. (2023a)): FIT, EXPLAIN, REFLECT, and REVISE, where the last three modules are the core modules of LSX (*cf.* Fig. 1). Let us now describe these four modules in more detail based on the pseudo-code in Alg. 1.

**Optimize for the base task** (FIT). The FIT module describes the underlying, base learning task in which the learner is optimized to solve a particular problem, *e.g.*, supervised image classification. Ultimately, FIT returns a model that has been optimized for the base task, *i.e.*, $f \leftarrow \text{FIT}(f, \bar{X})$ (*cf.* Alg. 1). In general, this module represents the vanilla learning approach for the underlying task. The details of this are thus independent of LSX. *E.g.*, for the task of supervised image classification a CNN-based learner might be optimized via a standard cross-entropy loss. More formally, $f$ is provided with a sample from the training dataset, $(x, y) \in \bar{X}$, and makes predictions $\hat{y} = f(x)$. Its latent representations are optimized given a corresponding loss function, $l_{\text{B}}$ (*e.g.*, cross-entropy: $l_{\text{B}} = l_{\text{CE}}(\hat{y}, y)$). Generally, within FIT, one can employ any bells and whistles of modern machine learning setups (*e.g.*, hyperparameter optimization via cross-validation, learning rate schedulers, etc.) and optimize it until it has reached satisfactory performances.

**Provide explanations** (Explain). Explain represents the first of three core modules of LSX. In this module, $f$ provides `explanations` to its predictions for a set of data samples[1], $\bar{X}_c \subseteq \bar{X}$. This is achieved via a pre-selected explanation method that returns an `explanation` for each sample in $\bar{X}_c$, *i.e.*, `explanations` $\leftarrow$ Explain$(f, \bar{X}_c)$ (*cf.* Alg. 1). The learner's explanation method can be selected from one of the vast collection of post-hoc or inherent eXplainable AI (XAI) approaches (*cf.* Guidotti et al. (2019); Ras et al. (2022); Liao & Varshney (2021); Carvalho et al. (2019)). Given the architectural constraints of the learner a suitable method can, *e.g.*, represent an input attribution method (that indicates how important a specific image pixel was for a model's prediction), a concept-level, logical statement (*e.g.*, `bluebird(X):- in(Obj,X),wingcolor(Obj,blue),size(Obj,small)`), or a generated natural language explanation (*e.g.*, "The person is playing tennis, because they are holding a tennis racket."). Lastly, we note that in certain cases it may be beneficial for $\bar{X}_c$ to represent a separate set that is with-held during the initial optimization of $f$, *i.e.*, $\bar{X}_c \cap \bar{X} = \emptyset$ (will be discussed in following sections).

**Provide feedback on quality of explanations** (Reflect). In the second core module, and arguably the most distinctive module of LSX, the high-level role of the internal critic is to "reflect" on the quality of the learner's explanations. This is an abstract measure and in LSX is quantified by the ability of the critic to perform the base task given the learner's explanations. This quantification is obtained in the Reflect module of LSX, *i.e.*, `feedback` $\leftarrow$ Reflect$(c, \bar{X}_c, $`explanations`$)$ (*cf.* Alg. 1). `feedback` hereby contains the critic's assessment (quantification) of the explanation's quality and represents a form of feedback for the learner (further discussed in the Revise module below). This can, *e.g.*, represent a probabilistic ranking over all `explanations` (how likely they are given the original data), but also a gradient signal from a classification loss function. The specific form of this feedback varies and depends on the model type of the critic, but also the type of provided explanations. We provide some concrete examples of this in our experimental evaluations. In any case, `feedback` must contain some form of information that can be utilized to revise the learner's representations based on the quality of its provided explanations.

By evaluating the quality of explanations based on their benefit in solving a task, the Reflect module represents one of the core aspects of LSX and is related to works concerning the evaluation and utility of AI model explanations (*e.g.*, Pruthi et al. (2022) and Fok & Weld (2023)). Moreover, in contrast to XIL where the corresponding "Obtain" module receives *human* explanatory feedback (*cf.* Friedrich et al. (2023a)) in LSX's Reflect module feedback is provided by a *non-human* model, *i.e.*, the critic, a second, internal AI model. In summary, Reflect returns a feedback signal that contains the critic's quality assessment of the learner's explanations.

**Integrate feedback on explanations** (Revise). In the last module of LSX, Revise, the `feedback` is utilized to refine, *i.e.*, finetune, the learner based on the quality of its explanations. Its goal is to ensure that the learner makes predictions based on "good" explanations. Therefore, the learner jointly optimizes its explanations based on the critic's feedback, but also based on its predictive performance for the original task. This prevents the learner from learning to provide useful explanations that are not actually used in its own decision processes.

The exact procedure of how to integrate the critic's feedback depends on the submodel types, but more importantly, on the form of the critic's feedback. It can be realized via one of the many ways provided in interactive learning settings (Teso & Kersting, 2019; Friedrich et al., 2023a), *e.g.*, loss-based methods (Ross et al., 2017; Selvaraju et al., 2019), but also non-differentiable revision approaches, *e.g.*, data augmentation approaches (Teso & Kersting, 2019) or retrieval-based setups (Friedrich et al., 2023b; Tandon et al., 2022). In the first case (*i.e.*, loss-based) an explanation loss, $l_{\text{expl}}$, is added to the base loss: $L = l_{\text{B}} + \lambda l_{\text{expl}}$ (where $\lambda \in \mathbb{R}$ represents a scaling factor). *E.g.*, $l_{\text{expl}}$ can represent a HINT-like loss (Selvaraju et al., 2019) that aligns the learner's explanation with the explanation that the critic has identified as "best". In the second case (*i.e.*, non-differentiable), the information stemming from `feedback` gets integrated into the overall learning setup, *e.g.*, by adding augmented training data samples or storing the revisory feedback in an explicit revision corpus. Overall, the result of the Revise module (*cf.* Alg. 1) is a finetuned learner model: $f \leftarrow$ Revise$(f, \bar{X}, $`feedback`$)$.

---

[1]It may be sufficient and more computationally feasible for the critic to only assess a subset of all data.

Table 1: A tabular overview of the different example LSX instantiations of our evaluations. The differentiation is based on different learner and critic models and the introduced LSX typology (FIT, EXPLAIN, REFLECT, REVISE).

| Instance | Learner | Critic | Input | FIT | EXPLAIN | REFLECT | REVISE |
|----------|---------|--------|-------|-----|---------|---------|--------|
| CNN-LSX | CNN | CNN | Image | Image Classification | Input Attribution | Explanation Classification | CE-loss on Input Attr. |
| NeSy-LSX | NeSy Concept Learner | Differentiable Reasoner | Image | Image Classification | Logical Statement | Explanation Ranking | MSE-loss on Concepts |
| VLM-LSX | VLM | LM | Image + Text | Language Generation | Textual Explanation | Explanation Ranking | CE-loss on Textual Expl. |

The last three modules (EXPLAIN, REFLECT, REVISE) can be repeated, *e.g.*, until an iteration budget $T \in \mathbb{N}$ is reached (*cf.* Alg. 1). Overall, the above typology presents the basic building blocks that an LSX instantiation should contain. As mentioned above, the choices of some components influences the choices of others. We further illustrate this via example instantiations in the context of our experimental evaluations.

## 3 Experimental Evaluations

In the following, we investigate the effects of Learning by Self-Explaining across various metrics, datasets and base models. To provide thorough examinations, we employ three different base models in LSX: a convolutional neural network (CNN), a neuro-symbolic model (NeSy), and a vision-language model (VLM). Based on these, we investigate the potential benefits of LSX concerning test-set generalization, explanation consolidation, explanation faithfulness and shortcut learning mitigation. We further provide extensive ablation evaluations, discussing the potential limits of LSX. Let us first specify our research questions followed by the model and experimental setup[2]. The investigated research questions are:

**(Q1)** Can training via LSX lead to competitive predictors? **(Q2)** Can training via LSX lead to improved generalization? **(Q3)** Can training via LSX help mitigate shortcut behavior? **(Q4)** Can training via LSX lead to more task-relevant explanations? **(Q5)** Can LSX lead to more faithful model explanations? **(Q6)** Can we observe predictive improvements via LSX in settings that go beyond one modality?

### 3.1 Example LSX Instantiations

We here provide a brief overview of the three investigated LSX instantiations. These represent examples of how to integrate a base model into LSX. Tab. 1 provides an overview of the instantiations based on our introduced typology. We briefly summarize these in the following and refer to Sec. A.1, Sec. A.2 and Sec. A.3, respectively, for full details.

**CNN-LSX.** For the CNN-based instantiation, both the learner and critic correspond to a CNN. This instantiation is evaluated in the context of supervised image classification wherefore in FIT the learner is optimized via a standard cross-entropy loss (CE-loss). The explanation method of the learner is an input attribution approach. Thus, the attribution-based explanation indicates the importance of each input pixel for a final class prediction. In the REFLECT module the critic attempts to classify the learner's explanations, *i.e.*, predict the image class of an explanation's underlying input sample. The quality of this *explanation classification* is measured via a second cross-entropy loss for the critic. Finally, in the REVISE module, the learner is finetuned based on a training signal from the learner classifying the original input images *and* the critic's (cross-entropy) classification loss over the provided explanations.

---

[2]Code available at: https://github.com/ml-research/learning-by-self-explaining

**NeSy-LSX.** In the NeSy-LSX example instantiation, the learner corresponds to a neuro-symbolic concept learner (Stammer et al., 2021; Koh et al., 2020) which extracts concept representations from raw images (*e.g.*, a bird's wing color) and makes final predictions based on the activations of these concepts for a given input image (*e.g.*, an image portrays a bluebird). The critic corresponds to a differentiable reasoner (Shindo et al., 2023; Rocktäschel & Riedel, 2017) which encodes explicit reasoning (in contrast to a CNN) thereby allowing to evaluate the likeliness of logic statements of a corresponding image. We evaluate NeSy-LSX in the context of image classification and optimize the learner via cross-entropy loss. The learner provides explanations for its class predictions in the form of logical statements, *e.g.*, image `X` corresponds to class `bluebird`, because it contains an object with a `blue` wingcolor and is of `small` size[3]. Hereby, the learner provides a set of candidate explanations per image class. In the REFLECT module, the critic performs a form of *explanation ranking*, where it estimates entailment probabilities for each candidate explanation via forward-chaining inference and given the corresponding input images. Based on these probabilities, the best explanation is selected per image class. Finally, in REVISE the learner is optimized for the original image classification task *and* for predicting the selected best explanation (via a mean-squared error (MSE) loss on the learner's concept-level representations).

**VLM-LSX.** For the VLM-based example instantiation, the learner corresponds to a pre-trained vision-language model (VLM). The critic, on the other hand, represents a pre-trained language model (LM). In contrast to previous instantiations, the input here consists of image and text. This way, we can evaluate LSX for visual question answering (VQA). To this end, the learner is optimized to generate textual answers to image-question pairs in line with ground truth answer texts via next-token prediction. The explanations in EXPLAIN are obtained by prompting the learner again with the image, question, its generated answer, and a query for generating an explanation. In more detail, the input prompt is *e.g.*, "{`image`}, the answer to {`question`} is {`generated_answer`}, because ...". The generated text corresponds to the model's explanation. As the language generation is auto-regressive and probabilistic, we sample a set of explanation candidates per input. Within the REFLECT module, the critic thus provides a preference (*i.e.*, *explanation ranking*) to each of these possible explanations. The ranking is based on the critic's (pretrained) knowledge and the corresponding input. Given these scores, the best explanation is selected. Finally, within REVISE, the learner is optimized jointly for generating an answer as well as generating the best explanation. In both cases this is done via a cross-entropy loss for next token prediction.

### 3.2 Experimental Setup

**Data.** To provide evidence for the benefits of LSX, we examine each instantiation via several suited datasets. Particularly, we examine i) CNN-LSX on the MNIST (LeCun et al., 1989) and ChestMNIST (Yang et al., 2023; Wang et al., 2017) datasets, ii) NeSy-LSX on the concept-based datasets CLEVR-Hans3 (Stammer et al., 2021) and a variant of Caltech-UCSD Birds-200-2011 dataset (Wah et al., 2011), CUB-10, and iii) VLM-LSX on the VQA-X dataset (Park et al., 2018). Furthermore, for investigating the effect of confounding factors (Q3), we also use the decoy version of CLEVR-Hans3, as well as DecoyMNIST (Ross et al., 2017) and ColorMNIST (Kim et al., 2019; Rieger et al., 2020). These three datasets contain confounded training and validation sets and non-confounded test sets. We note that in all CLEVR-Hans3 evaluations that do not target shortcut learning, the confounded validation set was used as held-out test set. We refer to Suppl. B for details on all datasets.

**Metrics.** We provide evaluations for LSX based on five metrics and briefly describe these here. **(1)** The first metric is the standard *classification accuracy* on a held-out test set. This is particularly used in the context of (Q1-3). **(2)** For investigating the revised explanations via LSX (Q4), we provide the classification accuracy of a linear, ridge regression model. This linear model is optimized to classify a set of explanations (given corresponding ground-truth class labels) and finally evaluated on a held-out set of explanations. The higher this model's accuracy, the more separable and distinct a learner's explanations. **(3)** For (Q4), we further provide a cluster analysis based metric over all explanations, similar to the Dunn index (Dunn, 1973; 1974). This metric, which we denote as *Inter- vs. Intraclass Explanation Similarity* (IIES), quantifies how similar explanations are within one class, but dissimilar between classes (lower values indicate better separability). For investigating whether the learner in fact makes a decision based on the reported explanations (Q5),

---

[3]In logical form, *e.g.*, `bluebird(X):- in(Obj,X),wingcolor(Obj,blue),size(Obj,small)`

Table 2: Improved (few-shot) generalization via LSX on various datasets and models. We here present the accuracy in % on a held-out test set across varying training set sizes.

|  | – MNIST – | | |
|---|---|---|---|
|  | 1.2k | 3k | full (60k) |
| CNN | 89.83±0.2 | 93.83±0.08 | **98.70**±0.1 |
| CNN-LSX | **91.59**±0.91 | **94.31**±0.43 | 98.03±0.2 |
|  | – ChestMNIST – | | |
|  | 1.6k | 4k | full (78k) |
| CNN | 58.68±0.15 | 58.49±0.31 | 60.86±0.08 |
| CNN-LSX | **61.16**±0.54 | **61.77**±0.75 | **63.41**±1.3 |
|  | – CLEVR-Hans3 – | | |
|  | 180 | 450 | full (9k) |
| NeSy | 91.40±1.80 | 96.81±0.94 | 99.00±0.28 |
| NeSy-LSX | **94.51**±1.94 | **97.34**±0.44 | **99.08**±0.17 |
|  | – CUB-10 – | | |
|  | 100 | 150 | full (300) |
| NeSy | 83.57±1.67 | 87.14±0.4 | 93.13±0.4 |
| NeSy-LSX | **84.49**±1.18 | **93.05**±1.72 | **96.33**±0.31 |
| avg. improvement | **2.07** | **2.55** | **1.29** |

Table 3: Mitigating confounders via LSX: Test set performances on confounded datasets with only confounded samples (*conf.*) and a subset of deconfounded samples during training (*deconf.*).

|  | – DecoyMNIST – | |
|---|---|---|
|  | conf. | deconf. |
| CNN | 63.52±1.39 | 86.88±0.68 |
| CNN-LSX | **78.99**±2.71 | **88.43**±2.34 |
|  | – CLEVR-Hans3 – | |
|  | conf. | deconf. |
| NeSy | 85.96±4.20 | 91.23±1.2 |
| NeSy-LSX | **90.90**±4.38 | **95.64**±2.21 |

we analyze the faithfulness (Hooker et al., 2019; Chan et al., 2022) of the learner's explanations via two metrics as introduced by DeYoung et al. (2020), namely **(4)** *sufficiency* and **(5)** *comprehensiveness*. Both metrics measure the impact of removing specific parts of the input (based on the model's explanations) on the model's predictive performance. Comprehensiveness measures the impact of *important* input features (as indicated by the model's explanations) on the model's performance. Sufficiency measures the impact of *unimportant* features. Both metrics provide a relative measure, whereby high comprehensiveness and low sufficiency score is better. We refer to Suppl. C for details of all described metrics.

**Setup.** In all evaluations, we compare the performances of each base model (CNN, NeSy and VLM), *i.e.*, when trained in its vanilla training setup, with the corresponding LSX-trained versions. When comparing these different training configurations, we use the same setup, *i.e.*, training steps, datasets, hyperparameters etc. We provide results as mean values with standard deviations over five runs with random seeds. For VLM-LSX we revert to one seed due to the resource demand of large-scale VLMs. We evaluate (Q1-5) in the context of CNN-LSX and NeSy-LSX and (Q6) in the context of VLM-LSX.

## 3.3 Experimental Results

**Improved (few-shot) generalisation (Q1-2).** We start by investigating LSX for improved generalization. To this end, we measure the held-out test set accuracy of CNN-LSX on the MNIST and ChestMNIST datasets and of NeSy-LSX on the CLEVR-Hans3 and CUB-10 datasets. Further, to evaluate for few-shot generalization, we use different-sized subsets of the original training set. The rightmost column of Tab. 2 shows test set accuracies when trained on the full training size of each dataset. We observe that on average (last row) there is a substantial improvement in accuracy —particularly for ChestMNIST and CUB-10. Thus, our results indicate that integrating explanations in a self-refining approach via LSX can lead to competitive, even improved performance. We therefore answer (Q1) affirmatively.

In the remaining columns of Tab. 2, we present the test-set accuracy in smaller-data regimes, *i.e.*, when the models were trained on different-sized subsets of the original training set. We observe large performance gains with LSX over all model configurations and datasets. Particularly, these improvements are greater than those observed on the full training set sizes. We provide important analyses and a discussion on the potential limits of this effect in the ablation evaluations of Sec. 3.4. Altogether, these results suggest that learning via self-explaining leads to improved test-set generalization performances and we therefore answer (Q2) affirmatively.

Table 4: Explanation consolidation via LSX. The metrics here are Inter- vs. Intraclass Explanation Similarity (IIES) of a learner's explanations (left) and the classification accuracy of a ridge regression model (RR. Acc., in %) on the learner's explanations (right). Both metrics are proxies for the explanation similarity within a class, yet separability between classes.

|  | IIES ($\downarrow$) | RR Acc. ($\uparrow$) |
|---|---|---|
| – MNIST – | | |
| CNN | $2.7\pm 0.07$ | $12.32\pm 0.35$ |
| CNN-LSX | $\mathbf{0.7}\pm 0.01$ | $\mathbf{99.91}\pm 0.06$ |
| – ChestMNIST – | | |
| CNN | $3.89\pm 0.13$ | $74.87\pm 0.24$ |
| CNN-LSX | $\mathbf{0.75}\pm 0.05$ | $\mathbf{99.92}\pm 0.03$ |
| – CLEVR-Hans3 – | | |
| NeSy | $0.65\pm 0.07$ | $93.48\pm 2.41$ |
| NeSy-LSX | $\mathbf{0.2}\pm 0.06$ | $\mathbf{100}\pm 0.0$ |
| – CUB-10 – | | |
| NeSy | $0.0266\pm 0.0005$ | $100\pm 0.0$ |
| NeSy-LSX | $\mathbf{0.0024}\pm 0.0001$ | $100\pm 0.0$ |

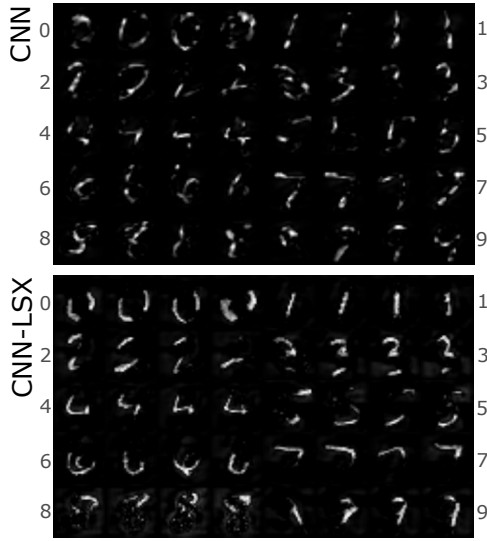

Figure 2: Exemplary explanations on MNIST from CNN baseline vs. CNN-LSX. Four random explanations are shown per image class (class ids on sides).

**Self-unconfounding (Q3).** In the second set of evaluations, we are interested in how LSX-trained models perform in the context of shortcut behavior (Geirhos et al., 2020). We particularly focus on confounded behavior as a form of shortcut learning in which a learner picks up on spurious correlations within the training dataset that are not present in the test set (Schramowski et al., 2020). We investigate two settings. (i) In the first setting, denoted as *deconf.*, $\bar{X}_c$ represents an explicitly deconfounded dataset that is with-held from training the learner (*i.e.*, $\bar{X}_c \cap \bar{X} = \emptyset$). Thus, the spurious correlation present in $\bar{X}$ is *not* present in $\bar{X}_c$ (*cf.* Suppl. B for details). (ii) In the second setting, denoted as *conf.*, we investigate the performance of LSX under the influence of confounding factors. In this case $\bar{X}_c \subseteq \bar{X}$ where $\bar{X}$ and $\bar{X}_c$ both contain confounded data samples. We note that the baseline models are trained on $\{\bar{X}, \bar{X}_c\}$. We focus on the Decoy-MNIST dataset for CNN-LSX and the original, confounded CLEVR-Hans3 version for NeSy-LSX.

In Tab. 3, we present the held-out test set accuracies of all configurations. We observe a strong improvement in test set performances when training via LSX on the deconfounded critic sets (*deconf.*). This suggests that reflecting on the explanations of an explicitly deconfounded critic set can lead to reduced confounding behavior over just adding these to the normal training set (as done in the baseline setup). Even more interesting, we observe that the LSX trained models in the *conf.* setting show greatly reduced confounding behavior in comparison to the baselines, despite both models having only ever seen confounded and never deconfounded data. We refer to Sec. 3.4 for ablations and a more detailed discussion on the observed effects. We additionally provide example explanations from both instantiations in Sec. C.3 which further support the findings of Tab. 3. Overall, our results show a remarkable beneficial effect of Learning by Self-Explaining in mitigating the issue of shortcut learning and we answer (Q3) positively.

**Consolidation of explanations (Q4).** In this evaluation, we wish to analyze how the critic's feedback signal influences the learner's representations, specifically its explanations. Based on the intuition behind the REFLECT module concerning "good" explanations, we hypothesize that the explanations of an LSX-trained model highlight distinct information which is more relevant for the underlying task, *e.g.*, wing color for distinguishing between bird species. We refer to such an effect as *explanation consolidation*. We base these evaluations on the Inter- vs. Intraclass Explanation Similarity[4] (IIES) and the accuracy of a ridge regression model that was trained to classify a set of explanations and evaluated on a second, held-out set. Both of these metrics measure the class-based separability of a model's explanations. Tab. 4 shows that training via LSX leads to much more separable and distinct explanations for both metrics. This effect appears less

---

[4]Note, that IIES is not normalized, thus only relative differences are important.

Table 5: Explanation faithfulness via LSX: Comprehensiveness and sufficiency results of explanations for models trained on all training samples.

| | Comp. ($\uparrow$) | Suff. ($\downarrow$) | | | Comp. ($\uparrow$) | Suff. ($\downarrow$) |
|---|---|---|---|---|---|---|
| | – MNIST – | | | | – CLEVR-Hans3 – | |
| CNN | $-1.34\pm0.39$ | $23.11\pm1.18$ | | NeSy | $59.3\pm3.6$ | $21.67\pm4.55$ |
| CNN-LSX | $\mathbf{16.49}\pm2.79$ | $\mathbf{-0.21}\pm4.18$ | | NeSy-LSX | $\mathbf{63.26}\pm2.57$ | $\mathbf{7.73}\pm1.28$ |
| | – ChestMNIST – | | | | – CUB-10 – | |
| CNN | $13.98\pm0.43$ | $-4.2\pm1.84$ | | NeSy | $64.54\pm0.2$ | $10.44\pm0.45$ |
| CNN-LSX | $\mathbf{18.84}\pm0.38$ | $\mathbf{-8.55}\pm1.92$ | | NeSy-LSX | $\mathbf{64.59}\pm0.23$ | $\mathbf{6.3}\pm0.34$ |

Table 6: Visual question answering accuracy (%) of a vision-language-based LSX instantiation (VLM-LSX) on the VQA dataset. $\text{VLM}_0$ represents the pretrained VLM and VLM (ft.) the VLM finetuned on the VQA dataset for question answering only and VLM-LSX the VLM finetuned via LSX.

| | $\text{VLM}_0$ | VLM (ft.) | VLM-LSX |
|---|---|---|---|
| | – VQAv2 – | | |
| VQA-Acc. (%) | 80.66 | 85.05 | $\mathbf{85.73}$ |

pronounced for the NeSy datasets, which is likely due to the sparseness and low dimensionality of their concept-level data and therefore also of the explanations. In Fig. 2, we also provide qualitative results of the explanation consolidation for MNIST, where explanations from four randomly sampled input samples are presented for each digit class. These visualizations support the quantitative results of Tab. 4 and particularly illustrate the distinctness of the explanations from an LSX-trained model. Overall, our results suggest that training via LSX results in more consistent explanations within a class and yet more distinct explanations across classes. Conclusively, we answer (Q4) positively.

**Explanation faithfulness (Q5).** In the next evaluations we investigate whether LSX-trained models produce explanations which are more faithful (Hooker et al., 2019; DeYoung et al., 2020; Schramowski et al., 2020). In other words, do the learners in fact use their explanations in their decisions. This is a relevant question as models that produce unfaithful explanations are detrimental *e.g.*, to building trust between human users and machines and at the same time make potential revisions via these explanations difficult (Schramowski et al., 2020; Friedrich et al., 2022; Teso et al., 2023).

We investigate the faithfulness of LSX-learned explanations with well-established metrics: *sufficiency* and *comprehensiveness* (DeYoung et al. (2020); *cf.* Suppl. C for details). In Tab. 5, we present the results of these metrics over the CNN and NeSy-based instantiations and the MNIST, ChestMNIST, CLEVR-Hans3 and CUB-10 datasets. One can observe a strong improvement via LSX in both metrics across all models and datasets. Specifically, the results show improved comprehensiveness for LSX-trained models. This indicates a higher dependency on important features of the explanation. At the same time, the LSX-trained models show improved sufficiency. Thus, unimportant information based on the explanations has a low impact on the learner's decisions. Overall, these results suggest that training via LSX also leads to more faithful explanations and we answer (Q5) affirmatively.

**Going beyond one modality (Q6).** Lastly, we provide results where we move beyond one input modality, but also beyond the task of supervised image classification. For this reason, we train a vision-language model (VLM) via LSX for the challenging task of visual question answering (VQA). Significantly, in this instantiation the model self-refines its *vision-language* representations only via language explanations. In Tab. 6, we provide test-set accuracies on the VQA dataset (Park et al., 2018; Lin et al., 2014) for three different training configurations. $\text{VLM}_0$ represents the off-the-shelf, pre-trained VLM (*i.e.*, zero-shot) and VLM (ft.) its VQA finetuned version, *i.e.*, finetuned only via FIT. Lastly, VLM-LSX is finetuned on the VQA task *and* via explanatory feedback from its language-model based critic. We observe that VLM-LSX leads to VQA performance improvements over the standard finetuned model. This depicts a ∼15% larger improvement over the zero-shot performance (*cf.* $\text{VLM}_0$) compared to the finetuned model. These preliminary results indicate that LSX potentially represents a beneficial learning approach outside of the

Table 7: Additional small data evaluations with 300 samples for MNIST and 30 for CUB-10.

| – MNIST (300) – | |
|---|---|
| CNN | $81.46{\pm}0.31$ |
| CNN-LSX | $\mathbf{81.80}{\pm}0.13$ |

| – CUB-10 (30) – | |
|---|---|
| NeSy | $\mathbf{66.02}{\pm}1.78$ |
| NeSy-LSX | $63.82{\pm}3.96$ |

Table 8: LSX via random critic feedback (w/ rand. $c$) on 3k MNIST and 150 CUB-10 training samples.

| – MNIST (3k) – | |
|---|---|
| CNN | $93.83{\pm}0.08$ |
| CNN-LSX | $\mathbf{94.31}{\pm}0.43$ |
| CNN-LSX (rand. $c$.) | $93.06{\pm}0.14$ |

| – CUB-10 (150) – | |
|---|---|
| NeSy | $87.14{\pm}0.4$ |
| NeSy-LSX | $\mathbf{93.05}{\pm}1.72$ |
| NeSy-LSX (rand. $c$.) | $87.24{\pm}2.93$ |

Table 9: Investigating confounded behavior on ColorMNIST.

| – ColorMNIST – | | |
|---|---|---|
| | conf. | deconf. |
| CNN | $\mathbf{59.34}{\pm}0.49$ | $\mathbf{87.24}{\pm}1.52$ |
| CNN-LSX | $57.62{\pm}0.49$ | $46.01{\pm}3.21$ |

realm of image classification. Yet, future investigations will be necessary to explore these findings in depth. Overall, we answer (Q6) positively.

### 3.4 Ablations

Our main results indicate promising benefits via LSX, however, wrong choices of individual components can, in general, lead to poor overall behavior. In the following, we therefore wish to investigate and discuss the limits and potential failure cases of LSX. We investigate three distinct cases that influence the performance of models trained via LSX. Notably, these investigations are not all-inclusive, but provide an important assessment for future endeavors. The first case focuses on the issue of *too* small data. The second case focuses on the issue of the critic's *capacity* and its influence on the learner's performance. Lastly, we investigate what happens when the explanation method is inadequate in describing discriminative features of the data in the context of confounding factors.

**Too small data.** With the first set of evaluations we wish to investigate the learner's performance in the case of more extreme data set sizes, particularly very small data sets. For this we provide further experiments with the example CNN-LSX and NeSy-LSX instantiations. We provide CNN-LSX with only 300 MNIST samples and NeSy-LSX with only 30 samples of CUB-10. The results are presented in Tab. 7 (test set prediction accuracies in %) and indeed suggest that the smaller the data set the less pronounced the effect becomes that we had originally observed in Tab. 2. Although we still observe improvements in the small data set regime with CNN-LSX, the results on CUB-10 in Tab. 7 even appear to suggest that given a very small data set LSX can have a negative impact on the learner's predictive performance. This difference is likely due to explanation selection in NeSy-LSX, *i.e.*, choosing the maximally probable explanation. If a bad explanation is chosen by the critic and enforced on the learner the learner's predictive performance will likely be negatively influenced by this (*cf.* Friedrich et al. (2023a)). Though it is difficult to adequately compare the two data set sizes (300 MNIST vs 30 CUB-10 samples) given the individual nature and complexity of them (grayscale images vs. concept-tagged images). Interestingly, the point on potentially "bad feedback" from the critic also hints at a form of overfitting that is quite specific to LSX (and other forms of explanation-guided learning): overfitted explanatory feedback can lead to poor predictive performances.

**Low capacity models.** In the second set of evaluations, we investigate the potential issues that arise from a critic submodel that performs poorly due to too low capacity, ultimately leading to random feedback. We evaluate CNN-LSX and NeSy-LSX where we have replaced their original critic submodels with simulated critic models that only perform random guessing on the base task. We denote this setting as "*rand. c*". In Tab. 8, we present the test set prediction accuracies of these training configurations on 3000 MNIST training samples for CNN-LSX and 150 CUB-10 samples for NeSy-LSX. Generally, we observe that indeed given a too-low capacity of the critic the learner falls back to its base predictive performance for both instantiations. These results seem to suggest a lower bound for LSX due to random explanatory feedback.

**Interplay between explanation method and confounders.** The inductive bias of the explanation method in combination with the nature of the confounding factor likely plays an important role in our observed effects on confounding mitigation in Tab. 3. Hence, we here investigate the influence of an "insuf-

ficient" explanation method. We adhere to investigating CNN-LSX's behavior on a third confounded data set, ColorMNIST (Kim et al., 2019; Rieger et al., 2020). Briefly, this data set represents another confounded version of MNIST. However, in contrast to DecoyMNIST, the confounding factor here is not spatially separate from the relevant features. In ColorMNIST, the pixels of each digit of a specific class are correlated with a specific color within the training set, but uncorrelated at test time. For example, all nines in the training set will possess a purple color, but random colors at test time (*cf.* Suppl. B). Thus, if a model has learned to focus only on the color of the digits it will incorrectly predict the class of a *purple* zero at test time to be the class "nine".

We consider an explanation method "insufficient" in the context of confounders if an explanation from a model that focuses on the relevant features (*e.g.*, the digit shape) is not distinguishable from an explanation from a model that focuses on the confounding features (*i.e.*, the digit color in ColorMNIST). Specifically, the explanation method utilized in CNN-LSX provides spatial input attribution maps that indicate which pixel of an input image is important for a prediction. However, a digit pixel could be important due to its color or position (*i.e.*, digit shape). In other words, in the context of ColorMNIST, such a type of (spatial) explanation is too ambiguous to distinguish relevant from confounding features. In this case, the explanation method of CNN-LSX is insufficient.

Tab. 9 presents the prediction accuracies on the non-confounded test set both of the vanilla trained CNN and the LSX-trained CNN for the two configurations *conf.* and *deconf.*. We observe quite different results from those of Tab. 3 (*i.e.*, those based on DecoyMNIST). Specifically, we observe confounding mitigation behavior neither for *conf.* nor for *deconf.*. On the contrary, we observe minor drops in accuracy for *conf.* and even larger drops in accuracy particularly in the *deconf.* condition. This suggests that when the explanation method does not adequately depict the right dimension that is needed for the data and task, learning via such insufficient explanations can even lead to a disadvantage for the overall model performance. We hypothesize that modifying the explanation method, *e.g.*, from an input-based to a concept-based explanation method, will likely undo this behavior. We hope future works will examine this.

## 4   Discussion & Limitations.

Overall, our evaluations provide evidence for the benefits of training via LSX on a variety of important tasks and metrics that go beyond standard evaluations of ML research. In the following, we wish to give a general perspective on LSX and provide a discussion of our findings and potential limitations.

**Configuration choices.** LSX can be instantiated in various ways, each with its own specific module and configuration choices. The example instantiations introduced in this work, however, already portray some interesting characteristics and differences which we wish to highlight here. For example, an important difference between CNN-LSX, NeSy-LSX and VLM-LSX lies within their REFLECT modules, specifically how the `feedback` of the learner's `explanations` is computed. In CNN-LSX, the `feedback` represents a differentiable signal, whereas in NeSy-LSX and VLM-LSX this represents a (probabilistic) ranking of a set of explanations. This second form of critiquing allows the model to weight the explanations and identify the most "useful" explanation. The first form of critiquing, on the other hand, allows the model to perform loss-based explanation fine-tuning. Related to this, but concerning EXPLAIN, where CNN-LSX utilizes *continuous* input-level explanations, the logical explanations in NeSy-LSX and the natural language explanations in VLM-LSX are *discrete*. As an effect of this, the form of revision in the REVISE module differs. In CNN-LSX one can simply pass the `feedback` from the critic to the learner via a backpropagated classification signal (*cf.* Tab. 1). In NeSy-LSX and VLM-LSX, however, we enforce the identified, most-likely explanation. Lastly, but also importantly, where in CNN-LSX and NeSy-LSX the critic evaluates the learner's `explanations` based only on the task information, the critic in VLM-LSX represents a pre-trained model that also utilizes its previously acquired knowledge (Zheng et al., 2023; Zhu et al., 2023; Li et al., 2024; Koo et al., 2024).

**LSX as explanation-based regularization.** Let us now take a different yet intuitive perspective on the effect of LSX. As previously stated, LSX combines ideas from self-refining AI and explanatory interactive learning (XIL). For XIL, several works (Ross et al., 2017; Selvaraju et al., 2019) have provided evidence that (human) explanations act as a regularizer during model refinement. We argue that this *explanation-based*

*regularization* plays a similar role in LSX, i.e. in the explanation-based, self-refining processes. Particularly, as our results on explanation consolidation show (*cf.* Tab. 4 in the context of Q3), the explanatory feedback from the internal critic leads to more task-relevant explanations. Furthermore, as our results on explanation faithfulness suggest (*cf.* Tab. 5 in the context of Q4) these explanations are consequently integrated into the model. We hypothesize that these guide the learner towards more distinct, task-relevant features which, in turn, benefits the model in terms of generalization (*cf.* Tab. 2 and Tab. 6). However, formal investigations will be necessary in future work to provide a theoretical justification of this intuition and our findings.

**Limitations.** Overall, although the results of Tab. 2 suggest benefits of LSX in terms of generalization, particularly in the context of small data, it is unlikely that this holds in all cases and instantiations. If either of the submodel's capacities are too low or the amount of data is too low (in contrast to the task and data complexity) it may not be possible for a learner to improve its overall predictive performance (*cf.* evaluations in Sec. 3.4). Moreover, if a learner's predictive performance after the initial FIT phase lies around random guessing this can greatly influence the quality of the explanations: as the model has not identified important features within the data it will not provide distinct explanations that reveal these. In the worst case, a resulting `explanation` can represent a distribution similar to uniform noise. Consequently, if the explanations are too bad and indistinguishable for the critic, it will likely not surpass random guessing such that the feedback to the learner will be uninformative. Furthermore, considering the additional evaluations of Tab. 9 it is important to conclude that LSX does not allow us to mitigate confounded behavior in all cases. Rather, it represents *one* potential tool in a toolbox of several mitigation strategies. In general, combating the diversity and complexity of different shortcut factors (Geirhos et al., 2020) most likely does not allow for a "one-size-fits-all" approach.

In addition, the embedded processing steps of the REFLECT module can lead to high computational costs for an LSX instantiation. Future work should, therefore, investigate more (computationally) optimal forms of reflecting and revising than, *e.g.*, introduced in CNN-LSX (*cf.* Sec. A.1). More generally, investigating potential challenges in scaling LSX to a wider range of machine learning applications (*e.g.*, scaling laws) will be necessary to solidify the findings of this work. Overall, our analyses and evaluations (particularly our ablation evaluations) suggest that it remains necessary for AI engineers to develop and assess the specifics of their LSX instantiations on a use-case basis. Thus, the notion of the no-free lunch theorem (Wolpert & Macready, 1997; Adam et al., 2019) also holds in the context of LSX.

Lastly, there remains a significant difference between human and AI-model self-reflection, which should be considered when deploying such agents. Irrespective of the human aspect and despite the promising results of our example instantiations there is still great potential for improvement for LSX, *e.g.*, via other design choices. Investigating such instantiations and their benefits is essential for consolidating the findings of this work. Specifically, providing a solid theoretical analysis of the limits of LSX, highlighted by our ablation evaluations, will be of great value in the future.

## 5   Related Works

**Explanations and Leveraging Explanations in ML.** Receiving explanations to an ML model's decision has become a heavily advocated and investigated topic in recent years, culminating in the field of *explainable AI* (XAI) (*cf.* (Guidotti et al., 2019; Ras et al., 2022; Roy et al., 2022; Saeed & Omlin, 2023)) and *interpretable AI* (Räuker et al., 2023; Li et al., 2018; Rudin et al., 2022; Rudin, 2019) with the later focusing on explicitly, *inherent* model explanations. Closely related to these fields are *self-explaining models* (Alvarez-Melis & Jaakkola, 2018; Lee et al., 2022; Roy et al., 2022; Camburu et al., 2018; Bastings et al., 2019; Majumder et al., 2022). Explanations from any of these approaches are used to evaluate the reasons for a model's decision. From backpropagation-based (Sundararajan et al., 2017; Ancona et al., 2018), to model distillation (Ribeiro et al., 2016) or prototype-based (Li et al., 2018) methods, the explanations often highlight important input elements (*e.g.*, pixels) for a model's prediction. However, several studies have also investigated multi-modal explanations (Rajani et al., 2020), logic-based explanations (Aditya et al., 2018; Rabold et al., 2019), and concept-based explanations (Stammer et al., 2021; Zhou et al., 2018; Ghorbani et al., 2019; Poeta et al., 2023). In all of these approaches explanations are only provided in a one-way communication as a means of model *inspection.*

The idea of leveraging explanations in a model's training process has only recently been picked up by parts of the ML community. *E.g.*, in explanatory interactive learning (XIL) (Teso & Kersting, 2019; Schramowski et al., 2020; Stammer et al., 2021; Friedrich et al., 2023a) human users provide revisory feedback on the explanations of an ML model. Similar ideas can also be found in other works of human-machine interactive learning (Teso et al., 2023; Gao et al., 2024; AlKhamissi et al., 2023), *e.g.*, in the context of imitation learning (Hu & Clune, 2023) but also in preference selection for aligning VLMs (Brack et al., 2023). Compared to these, we argue for the importance of leveraging explanations in the training loop even before the necessity of human-machine interactions. Indeed, several works have identified the value of leveraging explanations outside of human-interactive learning (*e.g.*, (Giunchiglia et al., 2022; Lampinen et al., 2022b;a; Norelli et al., 2022)). *E.g.*, in the works of Lei et al. (2016) and Bastings et al. (2019) (later categorized under the term *explain-then-predict models* by Zhang et al. (2021)), the goal is for one model to learn to extract the explanation from an input and a second model to learn to predict the final class from this. Similar ideas were picked up by (Zhang et al., 2021; Krishna et al., 2023). None of these works, however, evaluate the *usefulness* of explanations and particularly none use explanations as a means to revise a model.

**(Self-)Refinement in ML.** A recent, but quickly growing field of research related to our work is that which we categorize under the term of *self-refining ML*. This roughly encompasses research that investigates forms of self-supervised refinement of an ML model. *E.g.*, Wang et al. (2023) propose a self-refining approach for instruction-tuning. In the self-alignment approach of Sun et al. (2023), an LLM is aligned with few human provided principles. On the other hand, Schick et al. (2021) identify that LLMs can identify biases in their own generations and leverage this characteristic in a finetuning process to mitigate biased generation in future prompts. In the work of Madaan et al. (2023) a model is used to provide feedback to its own initial generations, where the feedback generation is guided via targeted, few-shot prompting. Zelikman et al. (2022), on the other hand, investigate finetuning a model based on generated "chain-of-thought" (Wei et al., 2022; Chung et al., 2022) rationales that lead to correct task predictions which is further related to the work of Shinn et al. (2023). Lastly, Paul et al. (2024) propose an approach in which a model learns to provide explicit intermediate reasoning steps for an answer via feedback from a critic model. In contrast to LSX, only a few of these approaches focus on refinement via explanations. Those that do require specifically developed modules for providing this feedback.

In contrast to self-refining ML, a separate branch of research focuses on revising a model based on feedback from a second, independent model (Du et al., 2023). *E.g.*, Such et al. (2020) introduce a meta-learning training data generation process in which a data generator and learner model are optimized to improve the learner's performance on a given task. Nair et al. (2023) propose an approach that leverages two agents, *researcher* and *decider*, to iteratively work through a task. In the student-teacher approach (Wang & Yoon, 2022) the goal is knowledge distillation, *i.e.*, learned knowledge from a trained model should be conveyed to a second student model. Interestingly, Pruthi et al. (2022) frame the utility of an explanation in a student-teacher setup in which the goal is for a student model to simulate a teacher's behavior. Also Schneider & Vlachos (2023) argue for the importance of explanations in reflective processes. However, in their proposed approach a model makes a final prediction based on the input and explanation that is estimated by a second model. Overall, many of these approaches have a different target and motivation than our work. Particularly, in LSX the role of the critic submodel is to provide explicit explanatory feedback to improve the learner's explanations and thus, indirectly, its predictive performance.

Overall, we consider many of these approaches to be complementary to our work and combining these with the ideas of LSX as very important future work, *e.g.*, combining *chain-of-thought* prompting (Wei et al., 2022) with LSX.

## 6 Conclusion

In this work, we have introduced a novel learning approach, Learning by Self-Explaining (LSX), with which we argue for a novel perspective on the role of self-explaining in the context of explainability and learning. With this approach, we claim that explanations are important not just for human users to understand or to revise an AI model. Rather, they can also play an important role in a form of self-refinement whereby a model assesses its own learned knowledge via its explanations. Our experimental evaluations highlight

several benefits of training via LSX in the context of generalization, knowledge consolidation, explanation faithfulness, and shortcut mitigation. Conclusively, with this work, we provide evidence for the potential of explanations within a model's (self)-learning process and as an important component for more *reflective* AI.

There are many avenues for future research related to LSX. Investigating LSX in the context of other modalities, *e.g.*, natural language, or other base tasks, *e.g.*, text generation, are both interesting possibilities. A more conceptual direction is the integration of a memory buffer of past LSX-refined explanations, thereby potentially allowing for models to re-iterate over previous explanations (Chi et al., 1994). Additionally, integrating background knowledge into the explanation reflection process presents an interesting direction whereby explanations are assessed not just based on the usefulness for the initial task, but also based on alignment with background knowledge constraints. Another important view is the connection between self- and causal explanations (Carloni et al., 2023; Zečević et al., 2021; Heskes et al., 2020), *e.g.*, can an AI agent utilize its self-explanations to perform interventional and counterfactual experiments (Woodward, 2005; Beckers, 2022)? Another crucial avenue going forward is to further apply LSX to other forms of supervision, such as self-supervised learning or reinforcement learning approaches, *e.g.*, via integration into actor-critic approaches or for guiding curiosity-driven replay (Kauvar et al., 2023). Lastly, we hypothesize that training inherently interpretable models via LSX (which provide more faithful explanations to their decisions from the start, *e.g.*, Delfosse et al. (2024)) could lead to improved effects than the ones observed here based on post-hoc explanation methods.

### Acknowledgments

This work benefited from the Hessian Ministry of Science and the Arts (HMWK) projects "The Third Wave of Artificial Intelligence - 3AI", "The Adaptive Mind" and Hessian.AI. It has further benefited from the "ML2MT" project from the Volkswagen Stiftung, the ICT-48 Network of AI Research Excellence Center "TAILOR" (EU Horizon 2020, GA No 952215), the Hessian research priority program LOEWE within the project "WhiteBox", and the EU-funded "TANGO" project (EU Horizon 2023, GA No 57100431) as well as from discussions as part of the DFG TRR 430 proposal "Expert-AI Cooperation in Natural Language". Furthermore, the authors thank Magdalena Wache and Alina Böhm for their preliminary results and insights on this research.

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

## Supplements

In the following, we provide details on the investigated LSX instantiations, experimental data, evaluation metrics etc..

# A    LSX Instantiation Details

We here provide details of the three example LSX instantiations that were utilized in our experimental evaluations. Exact implementation details can further be found in the corresponding code[5].

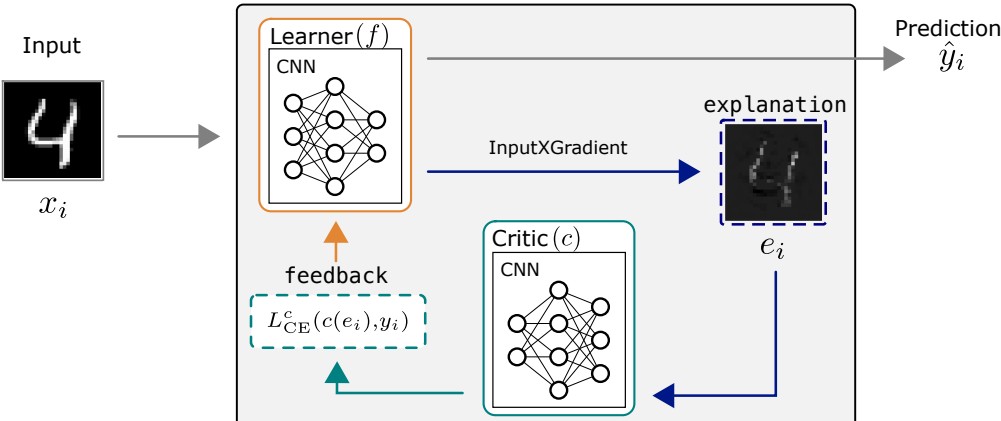

Figure 3: CNN-LSX: Learning by Self-Explaining instantiation for training CNNs for supervised image classification. Here CNNs represent both the *learner* and *critic*. Explanations are generated via InputXGradient. The `feedback` represents the classification loss of the critic on these explanations.

## A.1    CNN-LSX

Fig. 3 provides a graphical overview of the CNN-LSX example instantiation. We provide details based on this and Tab. 1 in the following.

**Learner.**

The learner corresponds to a convolutional neural network (CNN) with two convolutional layers, ReLU activation layers, one average pooling layer and two linear layers.

**Critic.**

The critic for CNN-LSX is identical to the architecture of the corresponding learner.

**Fit.**

Within the FIT module the learner is optimized via a cross-entropy loss for the task of supervised image classification with $l_{\mathrm{B}} := l^f_{\mathrm{CE}}(f(x_i), y_i)$ for $(x_i, y_i) \in \bar{X}$.

**Explain.**

The explanation method of CNN-LSX corresponds to the differentiable InputXGradient method described in Shrikumar et al. (2017) and Hechtlinger (2016) and implemented via the captum[6] pytorch package. Following the notation of Ancona et al. (2018), for an input sample $x_i$ and the output of model $f$ given the corresponding ground truth label, $y_i$, InputXGradient is defined as:

---

[5]Code available at: https://github.com/ml-research/learning-by-self-explaining
[6]captum.ai/

$$e_i = x_i \cdot \frac{\partial f_{y_i}(x_i)}{\partial x_i}. \tag{1}$$

This form of explanation method is considered a post-hoc explanation method as it estimates the explanation of a model after its decision processing (in comparison to inherent model explanations). Furthermore, it represents an input attribution method, *i.e.*, for each input element (*i.e.*, pixel) it provides an estimate of its importance for the model's final decision. Lastly, as Eq. 1 shows, InputXGradient contains the original input multiplied by the attribution information. Therefore, when the critic in CNN-LSX assesses these explanations the critic does not need additional information on the original input data, but receives this via the InputXGradient explanation.

**Reflect.**

In the REFLECT module for CNN-LSX the critic's task is to classify the explanations provided by the learner. Specifically, the critic is trained via a cross-entropy loss to predict the corresponding class of the learner's explanations. The critic trains for one epoch given the learner's `explanations` for the data of $\bar{X}_c$. We denote the set of these explanations formally as $E_c$. We allow the critic to update its parameters while iterating over all batches in $(E_c, Y_c)$ (with $Y_c \in \bar{X}_c$), whereby the loss values are accumulated over all batches and averaged. In practice we found that it was beneficial to reinitialize the critic with each LSX iteration. The final accumulated and averaged loss value is passed back to the learner and represents the `feedback` in CNN-LSX.

**Revise.**

In REVISE the learner performs the original base task for one epoch via $l_{\mathrm{B}}$ while jointly optimizing for the critic's explanatory `feedback` of the previous REFLECT module. Specifically, the learner optimizes a joint loss: $L = l_{\mathrm{B}} + \lambda l_{\mathrm{CE}}^c(c(e_i), y_i)$ for $(x_i, y_i) \in \bar{X}$ and $e_i$ based on Eq. 1. Hereby, $\lambda$ represents a scaling hyperparameter which we set high (*e.g.*, $\lambda \geq 100$) in our evaluations to prevent the learner from mainly optimizing for good prediction. Also here we refer to the corresponding code for the exact parameter values.

**Implementation details.**

In CNN-LSX we perform the EXPLAIN, REFLECT, and REVISE modules for several iterations until iteration budget $T$ is reached. In practice, we found it beneficial as a final step (*i.e.*, when the iteration budget has been reached) to perform a fine-tuning step in which we let the learner produce explanations for all samples in $\bar{X}$, $E^* = $ EXPLAIN$(f, \bar{X})$, and let $f$ be optimized for the base task making sure that it does not diverge its explanations from $E^*$ in the process. This is done via the combined loss $L = l_B + \lambda_{ft} l_{ft}(E', E^*)$, where $l_{ft}$ represents a simple mean-squared error loss between $E^*$ and the explanations that are generated via $f$ within each optimisation iteration. Lastly, we note that the CNN-LSX configurations are identical for all MNIST-based datasets.

For the results in Tab. 2 via CNN-LSX (for both MNIST and ChestMNIST) $\bar{X}_c$ presented about $\frac{1}{2}$, $\frac{2}{3}$ and $\frac{1}{2}$ of $\bar{X}$, from left column to right column, respectively. For the results in Tab. 3 via CNN-LSX on Decoy-MNIST we present the critic with 512 samples from approximately 60000 training samples for *w/ conf.* and 1024 test set samples for *w/ deconf.*.

**Computational details.**

This particular instantiation incorporates finetuning the critic on the learner's explanations, which represents a computational bottleneck. Specifically, processing one batch over the EXPLAIN, REFLECT and REVISE loop in our implementation takes $\approx 0.71$ seconds vs. $\approx 0.003$ seconds for the vanilla setup only via FIT. These results were averaged over ten batches. Notably, the vanilla setup is based on highly optimized pytorch code and refactoring of our implementations can lead to improvements. However, the major limitation remains, *i.e.*, finetuning the critic within the learner's finetuning step. It is thus important for future work to investigate more efficient instantiations, *e.g.*, based on retrieval-based feedback.

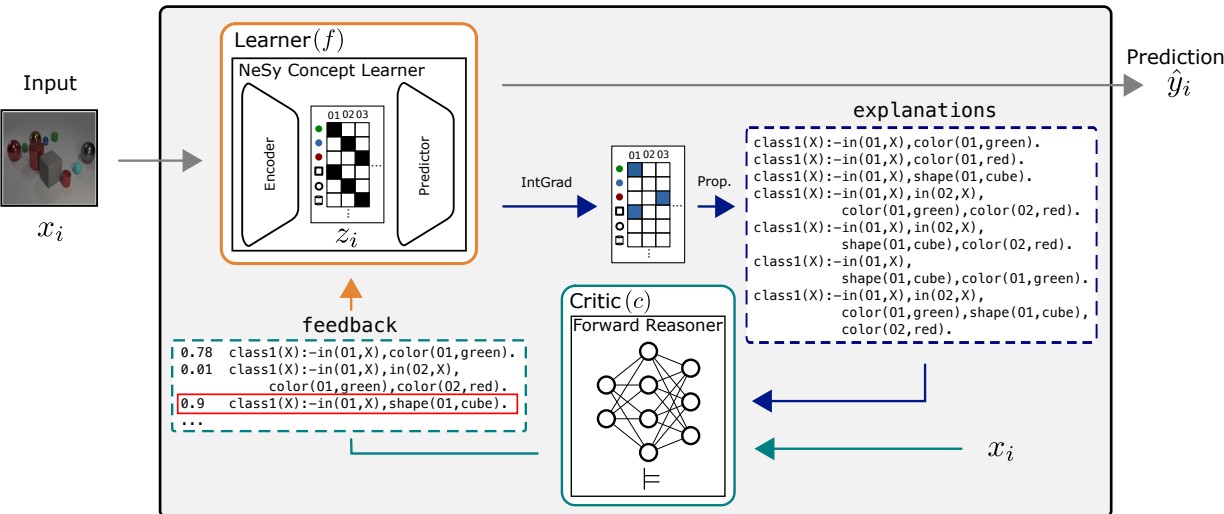

Figure 4: NeSy-LSX: Learning by Self-Explaining instantiation for supervised image classification via neuro-symbolic concept learner. The learner proposes a set of candidate class-specific logical explanations. The critic represents a neuro-symbolic forward reasoner, which computes the validity of these logical statements given visual input. The **feedback** represents a probabilistic ranking of the set of logical explanations with which we identify the most likely explanation per image class and revise the learner to only use this explanation for samples of that class.

## A.2 NeSy-LSX

Fig. 4 provides a graphical overview of the NeSy-LSX example instantiation. We provide details based on this and Tab. 1 in the following.

**Learner.**

The learner in the NeSy-LSX example instantiation belongs to a class of predictive models that we denote as neuro-symbolic concept learners. In our evaluations these consist of two submodules. The first module, the encoder, receives an image and transforms it into relevant concept representations. The second module, the predictor, makes the final task prediction based on the concept representations. Formally, an image, $x_i \in \bar{X}$, is first processed by a (pretrained) perception module into a symbolic representation, $z_i \in [0, 1]^{O \times A}$ which indicates the presence of objects and their attributes in the corresponding image. Here, $O$ indicates the number of objects and $A$ the number of predicted attributes. The reasoning module next makes a final task prediction given $z_i$.

The learner submodel for the NeSy-LSX instantiation differs for the CLEVR-Hans3 and CUB-10 datasets. For CLEVR-Hans3 the learner corresponds to the concept learner of Stammer et al. (2021) which incorporates a slot attention encoder for predicting the object's attributes (encoder) and a set transformer (predictor) for the final class prediction. As in Stammer et al. (2021), in our experimental evaluations, we make use of a pretrained slot attention encoder and perform updates only to the set transformer-based predictor model, *i.e.*, the module making the final predictions. For the CUB-10 configuration the learner corresponds to the setup of Koh et al. (2020) representing an Inception-v3 model Szegedy et al. (2016) for predicting the bird concepts (encoder) and a simple linear layer to make the final class prediction (predictor). Also here we perform updates only to the linear layer predictor model. The remaining LSX modules are identical over both datasets.

**Critic.**

The critic, $c$, of the NeSy-LSX instantiation corresponds to the neuro-symbolic forward reasoner of Shindo et al. (2023). Briefly, this critic evaluates how likely a logical explanation represents a logical statement that is present in a corresponding image. We refer to Shindo et al. (2023) for details. The predicate specifications etc. required for the forward reasoner for CLEVR-Hans3 correspond to the original ones

of Shindo et al. (2023). For CUB-10 we redefined each of the 28 bird concepts as neural predicates. *E.g.*, `haswingcolor` can take six different values which in the notation of Shindo et al. (2023) is defined as: `haswingcolor:brown,grey,yellow,black,white,buff`. We note that the forward reasoner treats missing values in a rule as zeros. Thus, the forward reasoner will only focus on those attributes that are provided with a rule, *e.g.*, the color green. This means that, *e.g.*, if a rule does not specify a shape constraint, the corresponding object can be of any shape. We refer here to our repository and the original work for details.

**Fit.**

Similar to the CNN-LSX instantiation the task of supervised image classification in the FIT module in NeSy-LSX is performed by optimizing via a cross-entropy loss $l_B = l_{CE}(f(x_i), y_i)$ with $(x_i, y_i) \in \bar{X}$. As previously mentioned, in our evaluations we hereby freeze the parameters of the encoder module of $f$, thus optimizing only the parameters of the predictor module of the learner.

**Explain.**

The EXPLAIN module of NeSy-LSX builds on the explanation approach of the concept learner of Stammer et al. (2021). Specifically, it first computes the integrated gradients (Sundararajan et al., 2017) of the symbolic representation, $z_i$, given the prediction. Following the notation of Ancona et al. (2018) for an input $z_i$ this is defined as:

$$\text{IntGrad}_i = (z_i - \bar{z}_i) \cdot \int_{\alpha=0}^{1} \frac{\partial f_{y_i}(\tilde{z}_i)}{\partial \tilde{z}_i}\Big|_{\tilde{z}_i = \bar{z}_i + \alpha(z_i - \bar{z})} d\alpha.$$

Hereby, $\bar{z}_i$ represents a "baseline" value, which in our evaluations corresponds to a zero vector. Next, the resulting attribution map on the latent concept representations, $e_{z_i} \in [0,1]^{O \times A}$, is binarized via a hyperparameter $\delta \in [0,1]$ to $e'_{z_i} \in \{0,1\}^{O \times A}$. We next transform these binarized attribution maps into logical statements which represent the final explanation form in NeSy-LSX (*cf.* Fig. 4). These logical statements consist of all subsets of conjunctive combinations of the important attributes and objects that are present in $e'_{z_i}$. We denote the set of these candidate logical explanations generated from sample $x_i$ as $\hat{E}_i$. For example, let us assume that for a specific CLEVR-Hans3 sample $x_i$ of class 1 we identify two objects to be important for the final class prediction. Hereby, the attributes *green color* and *cubical shape* are important for the first object and *red color* for the second object. Following the notation of Shindo et al. (2023) the set of generated candidate are:

```
class1(X):- in(O1,X),color(O1,green).
class1(X):- in(O1,X),shape(O1,cube).
class1(X):- in(O1,X),color(O1,red).
class1(X):- in(O1,X),color(O1,green),shape(O1,cube).
class1(X):- in(O1,X),in(O2,X),color(O1,red),shape(O2,cube).
class1(X):- in(O1,X),in(O2,X),color(O1,green),color(O2,red).
class1(X):- in(O1,X),in(O2,X),color(O1,red),color(O2,green),shape(O2,cube).
```

We refer to this step of constructing all potential candidate rules as propositionalizing (*i.e.*, changing the representation of relational data).

Notably, each input sample thereby produces a set of such candidate rules which may potentially contain many duplicates over samples of the same underlying class. Finally, by iterating over all samples in $X_c$, grouping the resulting candidate rules by image class and removing duplicates we receive a set of candidate rules per class as $\hat{E} = \{\hat{E}^1, ..., \hat{E}^K\}$, where $\hat{E}^k$ denotes the set of generated candidate logical explanations gathered over all samples of class $k$ and with duplicates removed.

For improved running times it is beneficial to limit the number of candidate rules per input sample by a maximum number of objects and attributes per object within an explanation rule *e.g.*, maximally four objects per rule. In our evaluations we set these two hyperparameters to still greatly overestimate the ground-truth rule and refer to the code for details (as well as for the values of $\delta$).

**Reflect.**

Having obtained the set of candidate `explanation` rules per image class, we pass these to the critic. For each underlying class and based on the data, $\bar{X}_c$, the critic next estimates the validity of each candidate rule, where we refer to Shindo et al. (2023) for details.

This evaluation is done for all positive examples of a class and for all negative examples of a class (*i.e.*, all remaining classes), resulting in two probabilities for the ith explanation candidate from set $\hat{E}^k$ of class $k$, which contains $L_k$ candidates in total. We denote these probabilities as $\rho^{k+} \in [0,1]^{L_k}$ and $\rho^{k-} \in [0,1]^{L_k}$, respectively. The first probability represents the validity of the rule as observed in samples only of the relevant class $k$ and the second represents the validity in samples of all other (irrelevant) classes ($j \in \{1, ..., K\} \backslash k$).

As we consider an explanation to be good if it distinguishes an input sample from samples of opposite classes, but indicates similarities to samples of the same class, we next compute the overall probability for each candidate logical explanation as $\rho^k = \rho^{k+} - \rho^{k-}$. The set of these probabilities over classes $P = \{\rho^1, ..., \rho^K\}$ represents the `feedback` of NeSy-LSX and represents the numerical values in the `feedback` representation in Fig. 4.

**Revise.**

Finally, per image class, $k$, we select the explanation rule with the maximal probability from $\rho^k$ corresponding to the red enclosed rule in Fig. 4. We denote this as $\hat{e}^k_{\max}$ with $\hat{E}_{\max} = \{\hat{e}^1_{\max}, ..., \hat{e}^K_{\max}\}$ for the set over all classes.

The selected logical explanations, $\hat{E}_{\max}$, are next mapped back into binary matrix form in the dimensions of the learner's latent symbolic representation $E'_{\max} = \{e'^1_{\max}, ..., e'^K_{\max}\}$ with $e'^j_{\max} \in \{0,1\}^{O \times A}$. This is required so we can compare, in a differentiable manner, the learner's explanations to the valid explanations as identified by the critic. Specifically, we compare the explanations in $E'_{\max}$ with those of the learner at the level of its symbolic representations, $e_{z_i}$ for $x_i \in \bar{X}$. Thus, in the REVISE module of NeSy-LSX we optimize a joint loss function corresponding to $L = l_B + \lambda l_{\mathrm{MSE}}(E'_{\max}, e_{z_i})$. For explanation $e_{z_i}$ of input sample $x_i \in \bar{X}$ with corresponding class label $y_i$ $l_{\mathrm{MSE}}$ is defined as:

$$l_{\mathrm{MSE}}(E'_{\max}, e_{z_i}) = \frac{1}{O \times A} \sum_{q=1}^{O \times A} (e_{z_{i_q}} - e'^{y_i}_{\max_q})^2.$$

In comparison to CNN-LSX in our evaluations we set $T = 1$ for NeSy-LSX. This means the critic only scores the proposed underlying logical explanations once and passes this back as feedback. Although it is in principle possible to perform multiple steps of this, *e.g.*, by first removing explanations which are most *unprobable* from the learner's representations and only after several of such iterations choose the most likely explanation, we leave this for future investigations.

**Implementation details.**

Note that for NeSy-LSX on CUB-10, we replaced the pretrained slot-attention module with a pretrained Inception-v3 network as encoder module (Szegedy et al., 2016) and the predictor module with a single linear layer as in (Koh et al., 2020). Hereby, the encoders where pretrained to predict the symbolic representations, $z$, *i.e.*, relevant object attributes. These modules were used in the NeSy configurations both for the baseline as well as LSX configurations.

In the setting of NeSy-LSX on CLEVR-Hans3 for the results in Tab. 2 $\bar{X}_c = \bar{X}$ for the two left columns and $\bar{X}_c = \bar{X} = \emptyset$ for the last column with $\bar{X}_c$ and $\bar{X}$ as 1500 to 7500, respectively (hereby, the critic was provided samples from the original training set of 9000 samples). For the results in Tab. 3 via NeSy-LSX on

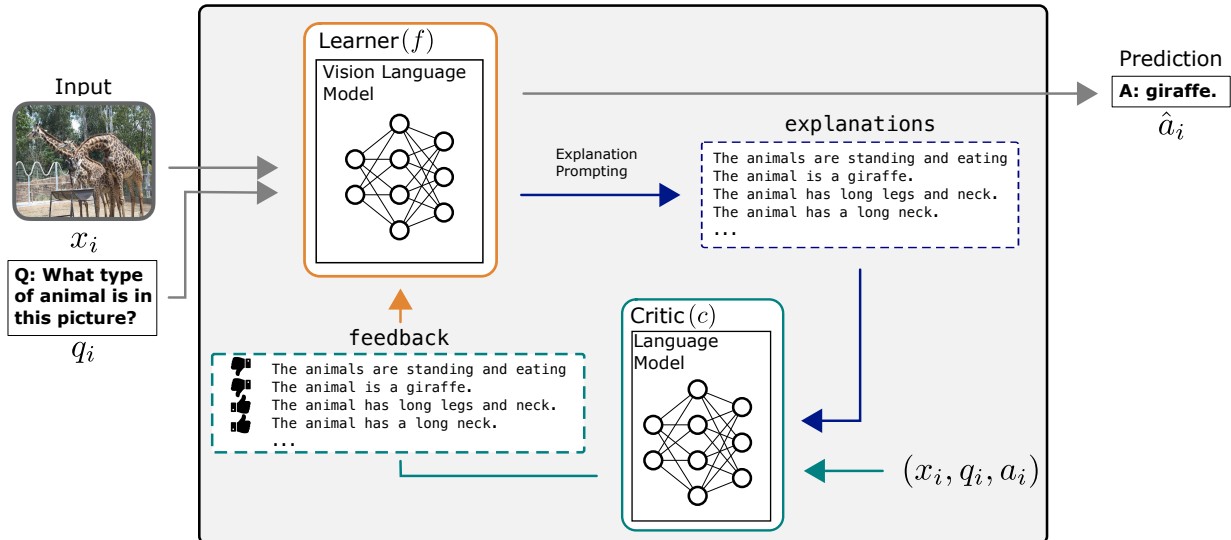

Figure 5: VLM-LSX: Learning by Self-Explaining instantiation for visual question answering via a vision-language model. The learner proposes a set of candidate **explanations** via explanation prompting. The critic represents a pre-trained language model, which provides preference scores over the generated explanations. The **feedback** represents a binary ranking of the set of explanations with which we identify "good" explanations and revise the learner for predicting these explanations.

CLEVR-Hans3 we present the critic with 50 samples and the learner 7500 separate training samples for *w/ conf.* and 150 test set samples for *w/ deconf.*

The setting of NeSy-LSX on CUB-10 for the results in Tab. 2 represented a small variation from the previous settings in that $\bar{X} \subseteq \bar{X}_c$, due to the small number of samples per class in CUB-10 in combination with the neuro-symbolic forward reasoner critic (requires sufficient amount of positive and negative class samples). In this case the datasets official validation set was concatenated with the (full) training set as a simple workaround. In this way for the results in Tab. 2 for CUB-10 from left column to right column the ratio between $\bar{X}_c$ and $\bar{X}$ was 29 to 24, 149 to 124 and 300 to 249, respectively. Note, the vanilla-trained models were also trained on the concatenated set.

### A.3 VLM-LSX

Fig. 5 provides a graphical overview of the VLM-LSX example instantiation. We provide details based on this and the overview in Tab. 1 in the following.

**Learner.**

The learner submodel of the VLM-LSX instantiation corresponds to the pretrained vision-language model MAGMA (Eichenberg et al., 2022). This VLM is built from a GPT-based LM backbone by adding a CLIP-based image prefix to encode images into the LM's embedding space. Subsequently, MAGMA trains dedicated bottleneck adapters on vision-language tasks to adjust the model to the new domains. Overall, in VLMs image and text can be input to the model as a joint sequence of tokens.

**Critic.**

We simulate a Language Model (LM) critic that provides preference scores for a set of textual explanations proposed by the learner. Specifically, we follow the methodology of Brack et al. (2023) and utilize a set of human-generated ground-truth explanations from VQA-X dataset (Park et al., 2018) as a form of knowledge base for the critic (further details in REFLECT below).

**Fit.**

The FIT module in the VLM-LSX instantiation represents the pretraining of the original MAGMA model. Specifically, Eichenberg et al. (2022) trained MAGMA on a collection of large-scale datasets for image captioning and visual question answering. Overall, with VLM-LSX we consider image-question-answer triplets $(i, q, a)$ consisting of an image, $i$, and a respective pair of text sequences for the question, $q$, and answer, $a$. This triplet is represented a sequence of tokens. In the FIT module the learner model is finetuned via such triplets. Let us denote $\bar{X} = ((I, Q), A)$ as the corresponding learner set, where in our evaluations of VLM-LSX $\bar{X} = \bar{X}_c$. The training of the VLM is finally performed via a cross-entropy loss for next token prediction of the answer. This is based on the next-token log-probability and conditioned on the previous sequence elements (we refer to Eichenberg et al. (2022) for details). This next-token loss represents $l_B = l_{\text{vqa}}(f((I, Q)), A)$ and optimization is performed via adapter-based finetuning (Houlsby et al., 2019).

**Explain.**

`Explanations` in VLM-LSX represent explicitly generated textual sequences. Specifically, we let the learner generate a set of $N_e \in \mathbb{N}$ explanations per sample, denoted as $E_i$ for sample triplet $(x_i, q_i, a_i) \in \bar{X}_c$. Here we follow the approach of Brack et al. (2023) for the explanation sampling process. Briefly this is based on a combination of top-k and temperature-based sampling and explanation prompt engineering, where an explanation prompt is the sequence of tokens appended to the image, question, and answer to elicit textual explanations. We select temperatures $T \in \{0.01, 0.1, 0.3, 0.6, 0.9\}$. This process overall results in $N_e = 25$ different natural language explanations per data sample.

**Reflect.**

Within the REFLECT model of VLM-LSX (and similar to NeSy-LSX) the critic provides preference `scores`, $\rho_i \in [0, 1]^{N_e}$, over the generated explanations. This preference scoring is based on calculating the sample-wise ROUGE-L score (Lin, 2004) between the learner's generated explanations and the annotated explanations that are stored in the critic's knowledge base. We further follow Brack et al. (2023) and consider every candidate with ROUGE-L $\geq 0.7$ as indicating a "good" explanation with respect to the ground-truth values.

**Revise.**

Based on the `feedback` of the previous REFLECT module we next select the `explanations` from $E_i$ with respect to the threshold defined above. We denote $E_{\max}$ as the set of these explanations over all samples within $\bar{X}_c$. We next add an additional loss to the learner's base loss: $L = l_B + l_{\text{expl}}(f((I_c, Q_c, A_c)), E_{\max})$ and optimize the learner via adapter-based finetuning (as in the FIT module) for predicting the corrext question answer and the best corresponding explanation.

**Implementation details.**

In our training setup we sample a total of $N_e = 25$ explanations, *i.e.*, 5 per temperature value. Furthermore, $\bar{X}_c = \bar{X}$. The number of LSX iterations was set to $T = 8$, where VLM (ft.) (*cf.* Tab. 6) was trained for the same number of overall steps. In the setting of VLM-LSX $\bar{X}_c = \bar{X}$.

## B  Data

**MNIST.** The MNIST dataset (LeCun et al., 1989) contains images of handwritten digits between 0-9. We utilize this dataset in the context of image classification, *i.e.*, an AI model should predict the underlying digit from an image. The number of training images in MNIST corresponds to 60k.

**ChestMNIST.** ChestMNIST (Yang et al., 2023; Wang et al., 2017) originates from the NIH-ChestXray14 dataset (Wang et al., 2017), which consists of 112,120 frontal-view X-ray images from 30,805 individual patients. Each image is originally associated with 14 disease labels extracted from text, making it a task of multi-label binary classification. In our setting we convert the data to a binary classification problem, *i.e.*, an image depicts an x-ray scan of a "normal" patient or "abnormal". The number of training images in ChestMNIST corresponds to 78k.

**DecoyMNIST** DecoyMNIST (Ross et al., 2017) represents a version of MNIST (LeCun et al., 1989) in which small boxes are placed randomly in one of the four corners for each sample (*cf.* Fig. 6). Importantly, the dataset contains confounders. Within the training set the gray boxes possess a specific grayscale value

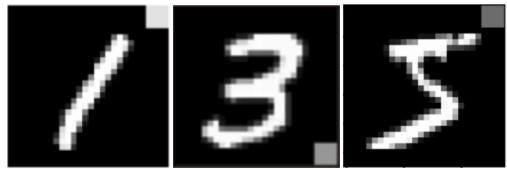

Figure 6: Example training images from DecoyMNIST.

for each digit class, where this grayscale value is randomized at test time. The number of training images in DecoyMNIST corresponds to 60k.

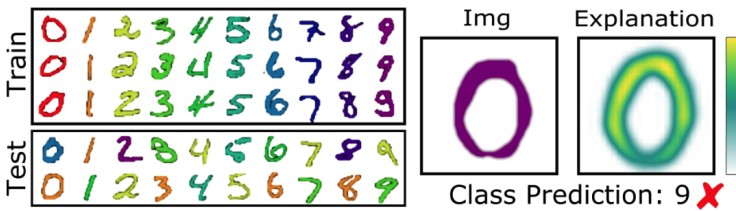

Figure 7: Example training images vs test images from ColorMNIST. Figure from Stammer et al. (2021)

**ColorMNIST** The ColorMNIST dataset (Kim et al., 2019; Rieger et al., 2020; Stammer et al., 2021) represents another confounded variation of the MNIST dataset. Hereby, the confounding feature is not spatially separate from the relevant features of the data. Specifically, in the training set each digit class is strongly correlated with a certain color. At test time however each digit is randomly assigned a color. (*cf.* Fig. 7). The number of training images in ColorMNIST corresponds to 60k.

**CUB-10.** CUB-10 represents a subset of the original Caltech-UCSD Birds-200-2011 dataset (CUB-200-2011) (Wah et al., 2011) that comprises images of the first 10 classes of the full dataset. Koh et al. (2020) originally perform a preprocessing step for CUB-200-2011 where concept vectors are replaced with the max voting vector over all samples of a class. In other words, the resulting concept activations are identical across all samples of a class which leads to a one-to-one mapping between concept activations and the class affiliation.

In CUB-10 we simulate a more realistic setting in which the class concept activations of (Koh et al., 2020) are overlaid with additional random noise, thereby maintaining the underlying class-based concept activation, but producing random variations per class sample. Specifically, we add uniformly distributed noise between 0 and 1 onto the class-based concept activations and binarize the resulting activations with a threshold of 0.75. The number of training images in CUB-10 corresponds to 300 images.

**CLEVR-Hans3** Fig. 8 presents the data distribution in CLEVR-Hans3 (Stammer et al., 2021). Specifically, CLEVR-Hans3 is based on the graphical environment of the original CLEVR dataset (Johnson et al., 2017), however reframed for image classification rather than visual question answering. Each class is represented by an underlying logical rule consisting of the presence of specific object combinations. Within the original training and validation set specific combinations of object properties are highly correlated, where they are not within the test set. *E.g.*, for the first class all large cubes are gray within the training set, but any color in the test set. This gray color thus represents a confounding factor within CLEVR-Hans3. The number of training images in CLEVR-Hans3 corresponds to 9k.

**VQA-X.** The VQA-X dataset (Park et al., 2018) extends the COCO (Lin et al., 2014) based VQA-v1 (Zitnick et al., 2016; Antol et al., 2015) and v2 (Goyal et al., 2017) datasets with human-annotated explanations. It contains open-ended queries to corresponding images which require a comprehension across vision, language, and common-sense domains for answering.

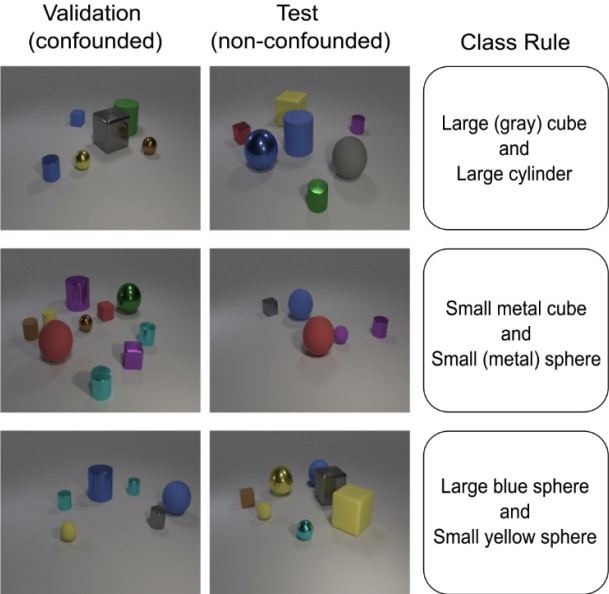

Figure 8: Data setup of the CLEVR-Hans3 dataset. Figure from Stammer et al. (2021).

## C  Metrics

### C.1  Explanation consolidation.

**Ridge regression classification.** For evaluating the separability of learned explanations, we provide the accuracy of a ridge regression (RR) model that is fitted on a set of tuples consisting of explanations from a trained learner and the corresponding ground-truth (GT) class labels of the underlying image. The RR model is fitted on a training set and tested on an additional, held-out set of explanations (and corresponding class labels).

This evaluation acts as a proxy of the separability of learned explanations. The higher the RR accuracy on the test set the better the separability between explanations. For each learning configuration in our evaluations we train a RR model separately on the explanations from the five differently seeded models.

**Encoding analysis.** The Inter- vs Intraclass Explanation Similarity (IIES) is defined as:

$$\text{IICS} = \frac{1}{K} \sum_{k}^{K} \frac{\frac{1}{M} \sum_{i}^{M} d(z_i^k, \mu^k)}{\frac{1}{K} \sum_{j,j \neq k}^{K} d(\mu^j, \mu^k)}$$

Essentially, this metric estimates in how far the explanations stemming from samples of one class are close to another compared to the explanations of samples from other classes. The encoding of a separate, pretrained model, $h$, provides the encoding space in which this similarity is assessed. The lower the values of IICS the better separable are the data for $h$. Specifically, $z_i^k$ corresponds to the encoding of the explanation of a sample $i$ from class $k$. This encoding is provided by an additional model, $h$, via $h(e_i) = z_i^k$, where $e_i$ is a provided explanation of sample $i$ from a learner $f$. $h$ is identical in architecture to the learner of which the explanations are being evaluated, however $h$ was separately pretrained only on the original task. For evaluating explanations from the CNN configurations, $h$ corresponds to an identical CNN that was trained for image classification as in the vanilla configurations. For evaluating the NeSy configurations a NeSy concept learner was pretrained for classification as in the NeSy vanilla setting. In both cases $h$ was provided with a random seed different from those used in the original training setups.

Furthermore, $\mu^k$ corresponds to the average encoding over all samples of class $k$ (where for notations sake we assume $M$ samples in each class, although this can vary in practice). $d(x, y)$ represents a distance metric between $x$ and $y$, where we have used the euclidean distance in our evaluations. We divide the distance

within one class by the average distance between the encoding mean of class $k$ and those of all other classes, corresponding to an estimate of the distance to all other class encodings. Finally this is averaged over all classes.

### C.2   Faithfulness

For comprehensiveness, parts of the input are removed that correspond to important features as identified by the explanation. As a result, the model should be less accurate in its predictions. In the case of sufficiency, one removes those input features which were deemed unimportant according to the explanation. Hereby, the model should not lose much accuracy. Notably, the original sufficiency and comprehensiveness metrics of (DeYoung et al., 2020) were introduced in the context of NLP in which input sequences are considered as discrete inputs. However, removing input features from continuous inputs such as images presents an issue (Hooker et al., 2019) as measured differences due to pixel removal may reflect the influence of the modified, out-of-distribution input rather than faithfulness of the explanation. For this case, we modified the metrics for the CNN configurations (*i.e.*, for explanations that are in a continuous form) to approximately compensate for this effect. For evaluating explanation faithfulness we thus provide results for CNN-LSX (and vanilla CNN) via the continuous adaptation of both metrics (denoted as $\mathrm{COMP}_{cont.}$ and $\mathrm{SUFF}_{cont.}$) and for NeSy-LSX (and NeSy vanilla) via the original comprehensiveness and sufficiency definitions (denoted as $\mathrm{COMP}_{discr.}$ and $\mathrm{SUFF}_{discr.}$). We formalize these in the following.

We follow the notation for $\mathrm{COMP}_{discr.}$ and $\mathrm{SUFF}_{discr.}$ of Chan et al. (2022). For this, $x$ denotes an input sample. We denote the predicted class of $x$ as $c(x)$, and the predicted probability corresponding to class $j$ as $p_j(x)$. Assuming an explanation is given, we denote denote the input containing only the $q\%$ important elements as $x_{:q\%}$. We denote the modified input sequence from which a token sub-sequence $x'$ are removed as $x \setminus x'$. Comprehensiveness and sufficiency for discrete explanations are finally defined as:

$$\mathrm{COMP_{discr.}} = \frac{1}{|B|} \sum_{q \in B} \frac{1}{N} \sum_{j=1}^{N} (p_{c(x_j)}(x_j) - p_{c(x_j)}(x_j \setminus x_{j:q\%}))$$

$$\mathrm{SUFF_{disc.}} = \frac{1}{|B|} \sum_{q \in B} \frac{1}{N} \sum_{j=1}^{N} (p_{c(x_j)}(x_j) - p_{c(x_j)}(x_{j:q\%})).$$

Where $N$ here represents the number of data samples in the evaluation set. In our evaluations we set $B = \{1, 5, 10, 20, 50\}$ as in the original work of DeYoung et al. (2020).

For computing comprehensiveness and sufficiency scores based on continuous explanations we first compute the comprehensiveness and sufficiency when a percentage $q$ of the top input elements (*e.g.*, pixels) are set to the median value of all input elements of the evaluation set. In comparison to the definition of $\mathrm{COMP}_{discr.}$ and $\mathrm{SUFF}_{discr.}$ of DeYoung et al. (2020) for the adaptation to continuous explanations we base the metrics on class accuracy rather than class probabilities. We denote these alternative computations as:

$$\mathrm{C\hat{O}MP_{cont.}} = \frac{1}{B} \sum_{q \in B} \mathrm{acc}(f(X \setminus X_{:q\%}^{\mathrm{median}}), y)$$

$$\mathrm{S\hat{U}FF_{cont.}} = \frac{1}{B} \sum_{q \in B} \mathrm{acc}(X_{:q\%}^{\mathrm{median}}, y).$$

Here, $\mathrm{acc}(f(X), y)$ corresponds to the accuracy score of a models prediction given input data, $f(X)$, compared to the ground truth labels, $y$. $X_{:q\%}$ corresponds to the full dataset in which everything but the top $q\%$ of

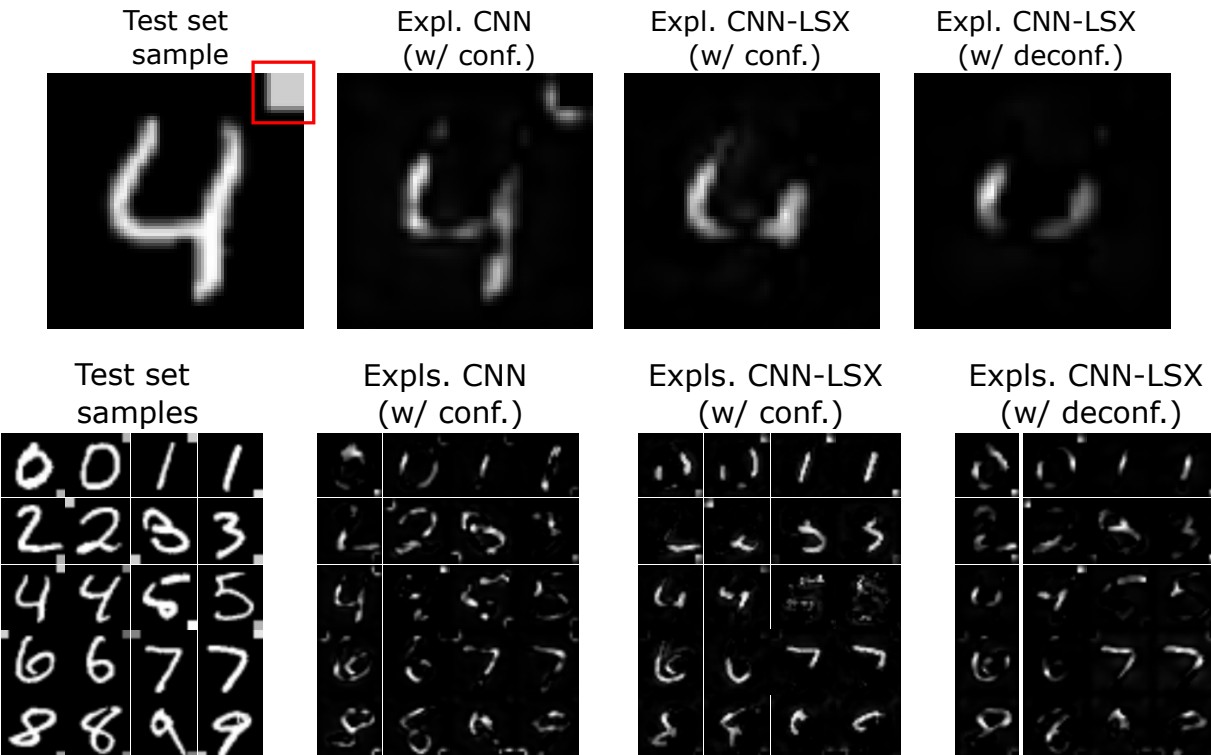

Figure 9: Example input attribution (InputXGradient) explanations from the different CNN configurations on DecoyMNIST. The top row show an original test set sample and the corresponding explanations for CNN (w/ conf.), CNN-LSX (w/ conf.) and CNN-LSX (w/ deconf.). The bottom row shows the same setup for 20 randomly selected test set samples. In red we have highlighted the confounding factor of the specific example. Note that the CNN-LSX models (both trained w/ conf. and w/ deconf.) do not indicate importance of the confounder in their explanations.

each samples input elements were set to the median value of the dataset and $X \backslash X_{:q\%}^{\text{median}}$ where the top $q\%$ of each samples input elements were set to the median value of the dataset.

Next we compute the same metrics, but when removing randomly chosen $q\%$ of the input elements by setting them to the median value. We denote these computations as $\hat{\text{COMP}}_{cont.}^{rand}$ and $\hat{\text{SUFF}}_{cont.}^{rand}$. Finally, we subtract these from the original values, leading to:

$$\text{COMP}_{\text{cont.}} = \hat{\text{COMP}}_{cont.}^{rand} - \hat{\text{COMP}}_{cont.}$$

and

$$\text{SUFF}_{\text{cont.}} = \hat{\text{COMP}}_{cont.}^{rand} - \hat{\text{COMP}}_{cont.}$$

## C.3 Self-unconfounding: Sample Explanations

Fig. 9 presents exemplary explanations from the different CNN configurations on the DecoyMNIST dataset. Specifically, we provide images from original test set samples (left), explanations of the baseline CNN (w/ conf.) (second to left), explanations of the CNN-LSX (w/ conf.) (second to right) and explanations of the CNN-LSX (w/ deconf.) (right). The explanations correspond to the InputXGradient importance maps. The top row represents images for a single sample, where the red box in the test sample image indicates the

confounder. We observe that both LSX configurations do not put any importance on this confounder for this sample. The bottom row shows the same setting for altogether 20 randomly selected test set images (two per class). Importantly, we observe a greatly reduced confounder importance in the explanations of the LSX configurations, though this is not fully removed (consistent with the accuracy results of Tab. 3).

Fig. 10 presents concept-level exemplary explanations ($e_{z_i}$) from the different NeSy configurations on class 1 images of the CLEVR-Hans3 dataset. Over four randomly chosen class 1 training images we observe that the baseline NeSy model puts great importance on the confounding factor of class 1 (*e.g.*, the gray color of large cubes, highlighted in red in the figure) the LSX based models both ignore this factor and even indicate the original groundtruth class rule (a large cube and large cylinder, highlighted in blue in the figure) despite never having received any explicit feedback on this. These qualitative results further indicate the confounding mitigation results observed in Tab. 3.

## D  Additional Discussion

**Human-machine interactions.** Accurate and trustworthy human-machine interactions have been identified as important criteria for the future deployability of AI systems (Friedrich et al., 2023a; Teso et al., 2023; Holzinger, 2021; Angerschmid et al., 2022). Also for LSX-trained models there is no guarantee that its explanations are aligned with human requirements and knowledge. This makes conclusive human assessment and potential human-based revisions necessary also for LSX trained models. However, in contrast to other learning frameworks, LSX directly facilitates the development and integration of such mechanisms that allow for fruitful human-machine interactions. *E.g.*, when integrating a model into LSX one must develop both the EXPLAIN and REVISE module. Via the EXPLAIN module a human user can directly query the learner's reasons for a prediction and via the REVISE module integrate feedback on these explanations. This can potentially ease the integration of necessary revisory feedback from humans.

**System 1 and 2 processing.** A prominent hypothesis from cognitive psychology (which has gained recent interest in AI research (Goyal & Bengio, 2022; Kautz, 2022; Ganapini et al., 2022; Booch et al., 2021)) is that human cognition can be described via two processing systems: an approximate, fast system (system 1) that handles the majority of familiar situations and an embedded, slower, yet more exact system (system 2) that processes unfamiliar settings (Kahneman, 2011). There are interesting parallels between this framework and that of LSX where FIT can be considered to represent a fast, initial processing phase, and the triad consisting of EXPLAIN, REFLECT and REVISE to represent a slower, embedded processing phase. An important open question, particularly in AI research on system 1 and 2 processing, is on the form of communication between the two systems (Goyal & Bengio, 2022; Kautz, 2022). Explanations, as utilized in LSX, possess interesting properties for this aspect. Specifically, explaining and reflecting on the learner's explanations in LSX represents a form of making the *implicit* knowledge of the learner *explicit*. At the same time, system 2 processing can also influence the processing of system 1 as alluded to by our findings on explanation consolidation (*cf.* Tab. 4). Lastly, our NeSy-LSX example instantiation has many parallels concerning the integration of neural and symbolic components to Henry Kautz's Neuro[Symbolic] system 1 and 2 approach (Kautz, 2022). Overall however, LSX is still far away from such models of human cognition and much additional research is needed.

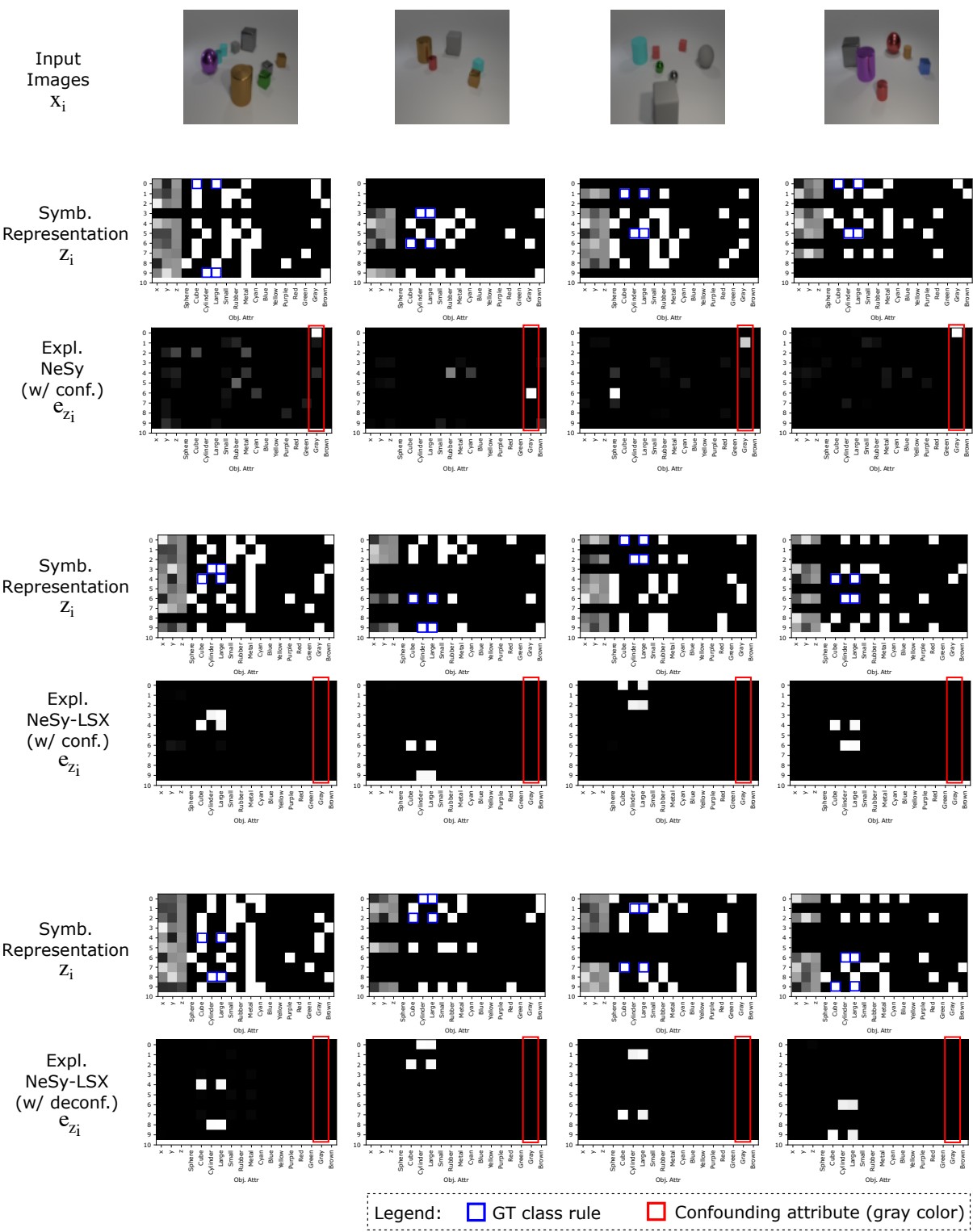

Figure 10: Example explanations from the different NeSy configurations for class 1 images of CLEVR-Hans3. Specifically, we provide images of the integrated gradients-based explanations, $e_{z_i}$. The first row depicts original images of four randomly selected training samples that belong to class 1. The second, fourth and sixth row depicts the symbolic representation, $z_i$, of these images, as processed by the slot-attention-based perception module, where row four and six merely represent row-wise permutations of $z_i$ in row two. Row three depicts explanations of baseline NeSy (w/ conf.). Row five depicts explanations from NeSy-LSX (w/ conf.) and the last row depicts explanations from NeSy-LSX (w/ deconf.). In red we highlight the confounding object attribute of class 1. In blue we highlight the underlying rule of class 1 based on each sample.

