# OpenReview forum: "Learning by Self-Explaining"
_TMLR — Accepted by TMLR_

### Review · Reviewer_fTZo · 2024-05-24

**Summary Of Contributions:**

The paper discusses the concept of "Learning by Self-Explaining (LSX)". The main idea behind LSX is that a learning module (referred to as the 'learner') performs a given task (e.g., image classification) and then provides explanations for its decisions. An internal "critic" module then assesses the quality of these explanations in relation to the original task. If the explanations are satisfactory, they can be used to refine the learning process. The authors have outlined four main components of LSX: Fit, Explain, Reflect, and Revise. The paper presents different implementations of LSX for two distinct learner models. The research finds that LSX can enhance the generalization abilities of AI models, reduce data requirements, and lead to more accurate and task-specific model explanations. In summary, there are 3 main contributions:
* Introduction of LSX: A new learning paradigm for machine learning that emphasizes self-refining by using explanations.
* Diverse Implementations: The paper showcases three instantiations of LSX, illustrating the flexibility of the sub-models and learning modules.
* Empirical Evidence: The research provides substantial experimental results on different datasets and metrics, highlighting the potential of LSX. They particularly note that LSX improves the generalization capabilities of base learning models, especially in scenarios with limited data. It also effectively addresses the influence of confounding factors, leading to clearer, task-specific model explanations.

---
I appreciate the efforts made by the authors to revise the paper and add experiments.

**Audience:**

Yes

**Broader Impact Concerns:**

There is no ethical concern as far as I know.

**Claims And Evidence:**

Yes

**Requested Changes:**

1. As mentioned in the weakness part, the introduction of VLM may lead to the requirements of using some recent work as baselines.
2. The notations a. and b. in Figure 1 are not obvious. Using a different color could be better.
3. It would be better to add a diagram in A.3 to describe the pipeline of VLM-LSX, similar to A.1 and A.2.
4. I feel that there are not many qualitative results in the experiments. Almost all results are shown in Tables except Figure 2. As a paper related to vision and language, it is not usual to have only two figures. I don’t have any concrete suggestions but more comparison figures with intuitive explanations will be more attractive to readers. This is not a mandatory request for changes.

**Strengths And Weaknesses:**

**Strengths:**

1.	The writing of the paper is much improved compared to the previous version. The structure of the paper is clearer after the reorganization.
2.	The new instances VLM-LSX looks very interesting and attractive. It makes this paper to have broader readers since the multi-modal capability of VLM extends this farmwork to other vision tasks.

**Weaknesses:**
1. VLM-LXS may raise some comparisons to recent works. Chain-of-thought and self-reflection [1], as the authors mentioned in the context, are two concepts that are very close to this work. In addition, I believe multi-agent debate [2] is also a related concept to this work (If we regard all agents as one system). So, I wonder if it is possible to use them as baselines in Table 6 to enhance their conclusion?
2. It seems that the code of VLM-LXS is not included in the code link.

---
[1] Shinn, N., Cassano, F., Gopinath, A., Narasimhan, K., & Yao, S. (2024). Reflexion: Language agents with verbal reinforcement learning. Advances in Neural Information Processing Systems, 36.

[2] Du, Y., Li, S., Torralba, A., Tenenbaum, J. B., & Mordatch, I. (2023). Improving factuality and reasoning in language models through multiagent debate. arXiv preprint arXiv:2305.14325.

---

> ### Author Response · Authors · 2024-07-03
>
> Thank you for your valuable time and feedback! We are happy to hear the structure and novel
> experiments have greatly improved our paper. Let us iterate over the remaining remarks concerning our
> resubmission.
>
> R: Comparison to other VLM works, chain-of-thought etc.
>
> A: We fully agree that the mentioned works [1,2] represent valuable approaches to finetuning, par-
> ticularly in the context of VLMs and LLMs. However, we consider these complementary to our approach
> rather than opposing. Specifically, where both works investigate ideas that split a task into several substeps,
> we here focus on the value of utilizing explanations within self-reflective processes. For now, we have not
> investigated the possibility of performing this on a subtask basis, but consider combining Chain-of-thought
> (CoT) with LSX a very important next step, yet out of the scope of this paper. Moreover, as shown in
> well-established recent studies [3,4] CoT prompting is not an approach that can be utilized out of the box
> on every model. E.g. , the size of the model can have an impact on its CoT abilities [3], but also whether
> it has been instruction tuned during its pre-training phase [4]. We consider LSX to represent a valuable
> approach in these cases where CoT might not be applicable. Overall, similar to what we had discussed in the
> context of shortcut learning, it is unlikely that one approach fits every problem setting. We, consider LSX
> to represent a potential tool in a toolbox of many approaches that facilitate improving a model’s reflective
> abilities. We have added the reviewer’s references to the related works section. Thank you for the hints.
>
> R: code of VLM-LSX
>
> A: We had provided a link in the original Readme concerning the VLM code. However, we have now
> made this more explicit now (created an explicit VLM-LSX folder).
>
> R: Notations in Fig. 1.
>
> A: We agree and have updated a and b to I and II and made the text bold. Let us know if these
> changes make the notations more obvious.
>
> R: Figure for A.3.
>
> A: We agree and have added a corresponding diagram in section A.3.
>
> R: Qualitative results
>
> A: We agree that qualitative results are helpful. For this reason we kindly refer to the appendix, e.g. ,
> Fig. 9 and Fig. 10 (in Sec. C.3 of the appendix). We have referenced these in the main paper in the context
> of our confounding evaluations (Q3).
>
> [1] Shinn, N., Cassano, F., Gopinath, A., Narasimhan, K., & Yao, S. (2024). Reflexion: Language agents
> with verbal reinforcement learning. Advances in Neural Information Processing Systems, 36.
>
> [2] Du, Y., Li, S., Torralba, A., Tenenbaum, J. B., & Mordatch, I. (2023). Improving factuality and reasoning
> in language models through multiagent debate. arXiv preprint arXiv:2305.14325.
>
> [3] Wei, Jason, et al. "Chain-of-thought prompting elicits reasoning in large language models." Advances in
> neural information processing systems 35 (2022): 24824-24837.
>
> [4] Chung, Hyung Won, et al. "Scaling instruction-finetuned language models." Journal of Machine Learning
> Research 25.70 (2024): 1-53.

---

### Review · Reviewer_SqW6 · 2024-05-25

**Summary Of Contributions:**

This paper proposes a new framework called Learning by Self-Explaining (LSX) where the learner improves its prediction via feedback from a critic. The implementation of critic and feedback can vary depending on the tasks, but overall they involve using an input attribution method as a regularizer. The paper proposes several instantiations of the proposed framework on tasks with different modalities and architectures, including CNN on MNIST, a neuro-symbolic model on CLEVR, and a pretrained VLM on VQA tasks. Experiments show that LSX can improve generalization performance, unconfound the predictions, and provide a more faithful explanation.

**Audience:**

Yes

**Broader Impact Concerns:**

There are no broader impact concerns in this paper.

**Claims And Evidence:**

No

**Requested Changes:**

- Computation complexity analysis (e.g., wall clock time).
- More experiments in the image classification setting. (more architectures, more datasets).

**Strengths And Weaknesses:**

**Strength**

The proposed framework seems interesting and well-motivated. The authors instantiated the proposed framework on a diverse set of problems to demonstrate its flexibility.


**Weakness**

The main criticism I have for the paper is that it's not very clear why the proposed method works and whether the method is scalable to most machine learning applications.

Let's start with the CNN on MNIST setting. In this setting, the explanation generated by `InputXGradient` and fed into the critic which in turn is used as a regularizer for the main model. First of all, this operation is extremely expensive since it effectively does an almost second-order update. This feels like it would be very expensive so I would like to see some complexity analysis and comparison of doing this. Furthermore, as I said, similar to self-training, it is not clearly why doing this would improve the performance since the explanation method is explaining the decision of an unregularized model. It is very interesting and potentially useful if doing this works, but to be convinced I would like to see larger models and more datasets (e.g., CIFAR or even Imagenet). Currently, only having the MNIST performance is not very compelling since one can easily get nearly perfect performance on MNIST with an MLP. There are plenty of pretrained CNNs available online so you would only need to fine-tune them

In NeSy-LSX and VLM-LSX, the critics leverage additional information -- NeSy-LSX uses an external neuro-symbolic forward reasoner and VLM-LSX uses human-generated ground-truth explanations. In my opinion, using additional information is not a particularly interesting approach and is more similar to distillation. More importantly, it makes hard to apply this method to other settings.

Overall, as a general framework for machine learning, I believe the paper would require more experimental evidence for the claims made and, if possible, some theoretical justification.

---

> ### Author Response · Authors · 2024-07-03
>
> We thank the reviewer for their valuable time and thoughts. We feel we need to clarify a few things,
> so let us dive right in.
>
> Actually, our overall claim is not to provide general instantiations that work out of the box for, e.g. every
> CNN setting. We fully agree that if this were our claim, additional experiments would be in accord and a
> theoretical analysis beneficial. Overall, LSX is described via the four general modules (Fit, Explain, Reflect,
> Revise) and not by our instantiations. This is analogous to the XIL framework [1,2].
>
> We particularly claim to exemplify on a diverse set of models and tasks the potential benefits of implementing
> this framework. Given our experimental evidence, we consider this claim to be well justified (as all other
> reviewers agree). In fact we had specifically followed the AE’s suggestion for resubmission to focus on
> restructuring our work and making our claims more clear rather than performing additional experiments.
> In light of this, we see that some phrases can be further improved. We have modified the wording of our
> RQs correspondingly and additional text throughout to better represent our underlying intended goals and
> claims. Overall, we fully agree that future work is required to establish our findings on a larger scale, but
> out of the scope of this paper.
>
> Furthermore, we are puzzled that the reviewer suggests we only have results on MNIST. Over the set of
> diverse example instantiations, we provide results on six very different datasets: CLEVR-Hans, MNIST,
> CUB-2011, Chest-MNIST, VQAV2 and Decoy-MNIST, whereby we specifically chose the instantiations to
> cover a broad range of current CV models (from small-scale to large-scale, from neural to neuro-symbolic,
> from CNNs to transformer-based architectures). Particularly we note that CNN-LSX was further evaluated
> on the complex, real world Chest-MNIST dataset.
>
> Overall, and analogous to the established XIL framework [1], the exact instantiation of LSX very much depends on an
> ML practitioner’s use case. Hereby, insufficient choices can lead to insufficient results, as we have illustrated
> in our ablations, e.g. InputXGradient explanations are insufficient in the ColorMNIST setting (a finding also
> identified in XIL research cf. [3]). In this context, we fully agree that the time complexity of our specific
> CNN-LSX instantiation can be greatly improved upon and consider it important future work to adapt this
> for other settings. In fact, we had already discussed this in the context of our ablations and limitations
> sections. We have now referenced a supplementary wall clock analyses (found in A.1) to underline this
> discussion; thank you for the hint.
>
> Lastly, we want to clarify that we investigate the benefits of our framework that go much more beyond
> simple accuracy. MNIST was hereby utilized as one of several settings for these investigations. While we
> do agree that in terms of pure classification accuracy, MNIST can be solved quite easily, our experiments
> particularly show benefits in terms of small data settings, explanation consolidation, explanation faithfulness,
> and confounding. Overall, in our investigations we evaluate LSX by comparing the performances of a
> deliberately standard model (e.g., ResNet-based CNN), when trained in the context of LSX, against when
> trained in the context of standard supervised training. The point of our evaluations is thus not to compare
> to all existing (MNIST-) baseline models which may provide higher accuracies.

---

> ### Author Response · Authors · 2024-07-03
>
> Further remarks:
>
> R: NeSy-LSX and VLM-LSX critics leverage additional information
>
> A: It is generally not necessary for any LSX instantiation to leverage additional knowledge. In par-
> ticular, the forward-reasoner-based critic of the NeSy-LSX instantiation does not have access to additional
> information. It evaluates the learner’s explanations via logical forward reasoning. However, it is not provided
> any additional information (other than that given to the learner) for doing this. Furthermore, human- anno-
> tated information is not necessary for a setting as in our VLM-LSX instantiation. We hereby have simulated
> an LLM via human-annotated information. However, in principle, this can be easily represented via the
> same LLM that is incorporated in the VLM-based learner. Overall, our evaluations aimed to illustrate a
> diverse set of example instantiations to provide valuable information for ML practitioners when developing
> their own specific instantiations. In light of this, we have investigated LSX for three important branches of
> current research: classical CNNs, recently emerging NeSy models, and recently emerging pre-trained large
> models. In this context, we find it very relevant to investigate the potential of LSX both when the critic is not
> provided additional knowledge and when the critic has additional knowledge, a setting that is particularly
> relevant given the current interest and surge of large pre-trained models (cf. RLHF, DPO etc). In addition
> we note that it is not required that the critic represents an identical version of the learner and it is up to
> the ML practitioner to choose an adequate critic module.
>
> R: “why the proposed method works”
>
> A: This is indeed an important question. We refer here to the relevant section in our discussions
> (“LSX as explanation-based regularization”) and and the ablations, where we discuss when LSX does provide
> beneficial effects. In general this discussion is connected to relevant findings from related XIL research [1,3]
> that identify that training via explanations serves as a valuable form of model regularization.
>
> [1] Felix Friedrich, Wolfgang Stammer, Patrick Schramowski, Kristian Kersting: A typology for exploring
> the mitigation of shortcut behaviour. Nature Machine Intelligence (2023)
>
> [2] Schramowski, Patrick, Wolfgang Stammer, Stefano Teso, Anna Brugger, Franziska Herbert, Xiaoting
> Shao, Hans-Georg Luigs, Anne-Katrin Mahlein, and Kristian Kersting. "Making deep neural networks right
> for the right scientific reasons by interacting with their explanations." Nature Machine Intelligence (2020)
>
> [3] Andrew Slavin Ross, Michael C. Hughes, Finale Doshi-Velez: Right for the Right Reasons: Training
> Differentiable Models by Constraining their Explanations. IJCAI (2017)

---

### Review · Reviewer_KDot · 2024-06-24

**Summary Of Contributions:**

I have reviewed the previous submission. So please refer to the various details in my previous review.

**Audience:**

Yes

**Claims And Evidence:**

Yes

**Requested Changes:**

Below I focus on the remaining points (or new points that are closely related to them).

In Table 3, NeSy-LSX improves the accuracy by about 5% under both conf and deconf data settings. Intuitively, I would expect that, if LSX improves the results mostly by deconfounding, then it won’t improve under deconf by much, as we see for the CNN-LSX. Would you have any good explanations for this?

In A.1, on the final step of Revise for CNN-LSX, I expect this final regularisation will have a strong effect (because it uses all training data), to the extent that I am not sure if this and the previous iterations are both necessary. Would you have strong intuitive reasons to do both? I suggest presenting ablations about the iterations and the final step. And, if the results do prefer using both, I am still curious why such reinforcement on training data won’t lead to overfitting; many sample images are used at least twice, and in the same way (extracting explanations).

In A.2, if I understand correctly, there is only one $e'\_{max}$ for each class, but there are multiple candidate rules for each sample image as to the previous page. Now, if your $e'\_{z}$ corresponds to a single rule, how do you select that rule? And, if $e’_{z}$ represents all the rules merged for an image, how do you merge the rules?

In A.2, for $e'\_{max}$ and $e'\_{z}$, how do you deal with missing values if an attribute is not mentioned in a (merged) rule? For example, the 1st line of your example rules doesn’t mention “shape”, so the shape value is missing.

### Minor

For the four “modules”, the authors at times also refer to them as “steps”. It is preferable to make the wording consistent, and I actually prefer “steps”.

Page3, “Where the learner performs an underlying base task”, typo, where → while

**Strengths And Weaknesses:**

In general, I am happy to see the paper improved greatly, and the additional experiments about shortcut mitigating/deconfounding are very interesting!

---

> ### Author Response · Authors · 2024-07-03
>
> We are happy to hear our revisions have greatly improved the paper and clarified open issues.
> Further, we are glad the reviewer is intrigued by our deconfounding experiments. We have fixed typos and
> focus on the remaining questions below.
>
> R: Table 3 intuition.
>
> A:
> Interesting question. When the critic in NeSy-LSX gets access to deconfounded data, the resulting rules will
> also not contain confounded information (the critic will select those rules that best describe the data it has
> been provided, i.e., data without confounding). Furthermore, given the discrete nature of the NeSy-LSX
> setting, the NeSy critic’s feedback is unambiguous in comparison to the CNN setting (there is no unique
> explanation e.g. for class 1 for CNN-LSX). Thus, the resulting rules potentially generalize easier to the
> test data. In summary, NeSy-LSX can make better use of the deconf. information than CNN-LSX, thereby
> explaining the larger improvement we had observed in table 3. It is important to note, however, that despite
> access to deconfounded data, all models do not reach fully deconfounded behavior (Tab. 2, 98% for MNIST
> and 99% for CLEVR).
>
> R: final step in A.1
>
> A:
> Overall, there are two practical reasons for the final Revise step: efficiency and stability. In the CNN-LSX
> instantiation the critic is untrained and in turn requires training. This training process is performed within
> the optimization iterations of the learner. To make these iterations efficient, we provided the critic only
> with a subset of data samples. Thus, during the main LSX iterations, the learner only receives regularizing
> explanatory feedback from the critic on this subset of samples. However, in the end we wish to utilise all
> information that is available in the full dataset, which we do in this last step. In addition, monitoring
> and tuning the learner’s optimization process (optimizing for the base loss and explanation loss) while
> simultaneously training the critic can be challenging in practice. We thus explicitly exclude the critic from
> the final optimization step, thereby making it easier to guarantee a good optimization of the learner based
> on both losses. Thus, in theory, the final Revise step (as performed for our CNN-LSX instantiation) is not
> necessary but we found it to be greatly beneficial in practice.
>
> Concerning the question regarding overfitting: to clarify, the learner in LSX is not provided more data
> than, e.g. in the baseline setup. Generally, overfitting is a problem that is not specific to training via LSX.
> Specifically, for a fair comparison, we had performed equally many learning steps in both setups (which did
> not lead to overfitting). Furthermore, as our evaluations show, the explanatory feedback in LSX serves as a
> form of regularization that helps to mitigate overfitting (e.g. as described in section “LSX as explanation-
> based regularization.”). As we have observed in our Q4 experiments, the learner’s explanations (given the
> critic’s feedback) become more class-specific and less overfitted to specific samples (e.g. depicted in Fig.
> 2 and Tab. 4). When the learner is then encouraged to incorporate such explanations within its decision
> processes it is regularized to focus on the more class-specific features that are represented in the explanation.
>
> R: single rule in A.2
>
> A:
> Correct, we first collect the candidate rules for each sample image (Ê_i for sample i). We then group all explanations that stem from samples of a specific class. After removing potential duplicates that occur from multiple samples of a class, we thus have a large set of candidate rules per class (Ê^k for class k). From this large set, we then select one most probable rule (e^k_{max}) based on the critic's scoring. We refer here to the second to last paragraph on page 25 of the pdf.
>
> R: missing values in A.2
>
> A:
> That’s an important point to clarify. In the critic step of NeSy-LSX (the neuro-symbolic forward
> reasoner), we treat “missing values” as zeros. This means that if a rule does not specify a shape constraint,
> the object can have any shape when the forward reasoner evaluates the rule. The forward reasoner will only
> focus on those attributes that are provided, e.g. the color green. This follows the initial implementation of
> Shindo et al. [1]. We have added this information in section A.2.
>
> R: consistent naming of modules
>
> A: Yes that is a good point and we have updated the naming accordingly. We specifically decided for
> "module" rather than "step" to be more consistent with the terminology of the XIL framework [2]. We hope
> the reviewer can agree on this.
>
> [1] Shindo, Hikaru, Viktor Pfanschilling, Devendra Singh Dhami, and Kristian Kersting. "αILP: thinking
> visual scenes as differentiable logic programs." Machine Learning (2023)
>
> [2] Felix Friedrich, Wolfgang Stammer, Patrick Schramowski, Kristian Kersting: A typology for exploring
> the mitigation of shortcut behaviour. Nature Machine Intelligence (2023)

---

> > ### Comment · Reviewer_KDot · 2024-07-22
> > **A late reply**
> >
> > I am sorry for the late reply. I hope the following will be considered.
> >
> > **Table 3.**
> > You explained why you think NeSy-LSX improved *more than CNN-LSX* under deconf data. But this does not explain why NeSy-LSX improved mostly *the same as under confounded data*. Again, if you believe NeSy-LSX improved mostly by deconfounding under deconf data, then, because the information of deconfounding is missing under confounded data, the improvement under confounded data is probably by extracting some information that is *not* contained in deconf data. But I cannot think of what kind of info it could be and how could LSX extract it.
> >
> > **Final step in A.1.** I can understand your practical considerations. However, my concern about overfitting was "many sample images are used at least twice, and in the same way (extracting explanations)" but not "the learner in LSX is [...] provided more data than, e.g. in the baseline setup". How large is the subset of training data you provided to the critic?
> >
> > **Single rule in A.2.** My question was how you form the $e_{z_i}$ on page 26.

---

> > > ### Author Response · Authors · 2024-07-24
> > > **Thanks for follow up!**
> > >
> > > Thanks for taking the time for your follow up questions!
> > >
> > > **Remarks on Table 3:**
> > >
> > > We apologise if we had misunderstood the initial remark. Indeed the NeSy critic of NeSy-LSX contains
> > > an inherent bias to favor shorter rules over complex rules, e.g. , "Large & Cube" rather than "Large & Cube
> > > & Gray" (where gray is the confounding factor). This bias thus leads also in the deconf case to the critic
> > > selecting non-confounded rules on average. This can explain the improvement that we observe in the conf
> > > case. We note however that this is less reliable in the conf case than in the deconf case as can be seen via
> > > the higher performance variance.
> > >
> > > Concerning the "same" amount of improvement in both the deconf and conf case for NeSy-LSX over NeSy:
> > > despite the absolute improvement of LSX over the baseline of ≈ 5% we note that the relative improvement is
> > > not identical over the conf and deconf settings. Specifically, in the conf case LSX removes one third percent
> > > of the baseline’s errors, in the deconf case we observe a relative reduction of 50% of the baseline errors. Thus,
> > > when the model is provided deconfounded critic samples we do indeed observe a larger overall improvement
> > > than when only provided confounded data samples.
> > >
> > > **Sample images used twice in final step in A.1.**
> > >
> > > Unfortunately, we are yet unsure if we understand "many sample images are used twice".
> > > If the remark refers to that samples are used both in the initial set of iterations as well as the final step, it is partly correct. Within the iterations, the samples are used the same way, \ie they are used for training the learner and then for explanation optimization through the critic. In that regard, the iterations could be considered as epochs.
> > > Afterwards, the samples are used in the final step to optimize the classification of the learner. However, the critic is not part of this step and at this point, the model explanations are only used for regularization (so that the explanations do not diverge during this final step).
> > >
> > > So overall, the samples are used multiple times during the initial iterations and then in the final step (without the critic), which can be understood as "epochs" of the total system.
> > >
> > > Subset sizes: we refer here to the bottom of page 23 for CNN-LSX (briefly: between 1/2 and 2/3 of the full dataset, depending on the initial size of the full dataset).
> > >
> > > If this information does not yet resolve the issue, we would be grateful if the reviewer could provide a more detailed explanation of where exactly in the framework they consider "samples to be used twice (in the same way)".
> > >
> > >
> > > **Follow-up concerning e_z_i:**
> > >
> > > Sorry for the confusion here. To further clarify the original remark: e'_z represents one rule that is
> > > obtained for each image based on the integrated gradients method (the blue box betwee the (the blue box betwee the IntGrad and Prop. arrow in Fig. 4).
> > > Specifically, e_z is the explanation from the learner for sample x_i, i.e., the integrated gradient explanation of that sample. e'_z is then the binarized representation of this explanation. Note that this binarized, tabular representation corresponds to a logical explanation rule. This is not a merged rule as the reviewer suggests. Rather, it can be a potentially long and therefore specific rule of this data sample. To increase the space of candidate rules for the critic to evaluate, we extract the possible subsets of conjunctive rules within this original rule, thus leading to \hat{E}_i, i.e., the set of these rules for sample x_i (via e_z_i). We refer to this process as propositionalisation which essentially corresponds to an enumeration of each conjunction within the original rule. E.g., \hat{E}_i is represented in the "explanations" (dashed blue) box in Fig. 4.
> > > We further refer here to the explain section on page 25.
> > >
> > > Now, once we have selected the best class-level explanation in the reflect module, e'^j_max (described in our previous response and further in the reflect and revise sections on page 26), within the revise module we compare the local explanations of each sample, e_z_i, to the relevant class-level explanation, e'^j_max, that was selected based on the critic.
> > >
> > > We hope this finally clarifies the question.
> > >
> > > Overall, we hope we could clarify all follow up remarks. Please do let us know otherwise.
> > >
> > > Best,
> > > the Authors

---

### Author Response · Authors · 2024-07-03
**Thank you for your valuable time and feedback**

We wish to thank all reviewers for their valuable time and their appreciation for our work. We are happy to hear that our additional experiments and restructuring have greatly helped to clarify and support our claims. The reviewers have particularly highlighted the diversity of our evaluations both in terms of datasets and exemplified models. We provide individual remarks below.

---

### Decision · Action_Editor_vRNM · 2024-08-13

**Recommendation:** Accept with minor revision

**Comment:**

### 1. Background
This is a resubmission, and I served as the AE for the previous version. Several reviewers have also participated in the previous round of review. I noticed that the authors have made significant revisions compared to the previous version. The following review focuses on the remaining issue identified in this round.

### 2. Outstanding concern on the supporting evidence
Some reviewers still feel the paper is too general, lacks clarity on why the proposed method works, and whether it is scalable to most machine learning applications. Therefore, they suggest that the paper requires more experimental evidence for the claims made and, if possible, some theoretical justification.

### 3. Acceptance criteria
According to the acceptance criteria, the claims made in the submission should be supported by accurate, convincing, and clear evidence. Any gap between claims and evidence should be addressed by the authors. Often, this will lead reviewers to ask the authors to provide more evidence by running more experiments. However, another way to address such concerns is for the authors to adjust (reduce) their claims.

### 4. Discussions

- **Concerns about "this paper is too general``**

The concern about the scope and specificity may come from the inherent nature of the research direction rather than the narrative presented in the paper. For instance, in the well-known Mixup paper (“Mixup: Beyond Empirical Risk Minimization,” ICLR 2018), similar concerns might be raised because *beyond ERM* is quite a broad concept in machine learning. Similar issues also frequently occur in the AI fairness and explainability papers. In these domains, different viewpoints in the research might lead to perceptions of over-claiming contributions.

Therefore the role of the AE in this context is to discern whether this concern arises from:
- *The narrative in the paper (or over-claiming).* If the issue lies here, major revisions should be required.
- *Inherent concerns about the research direction.* If the concern is related to the general nature of this research direction, it may reflect an inherent bias and could be addressed by discussing different viewpoints in the conclusion section.

Let’s have a look at the abstract

> Current AI research mainly treats explanations as a means for model inspection. Yet, this neglects findings from human psychology that describe the benefit of self-explanations in an agent’s learning process. Motivated by this, **we introduce a novel approach in the context of image classification, termed Learning by Self-Explaining (LSX)**.……

> We provide an overview of important components of LSX and, based on this, perform extensive experimental evaluations via three different example instantiations. Our results indicate improvements via Learning by Self-Explaining on several levels: in terms of model generalization, reducing the influence of confounding factors, and providing more task-relevant and faithful model explanations. Overall, our work provides evidence for the potential of self-explaining within the learning phase of an AI model.

I have read the revised paper again. Indeed the abstract stated

(a) *a novel approach LSX in the context of image classification*. In the paper, the corresponding experiments further considered three concrete image classification tasks.

(b) *overview of important components of LSX*. In the paper, this is illustrated.

(c) *model generalization, reducing the influence of confounding factors, and providing more task-relevant and faithful model explanations.* The experiments demonstrated this.

Based on these, this AE feels the paper’s actual contribution aligns the scope in the abstract. There is no significant overclaiming issue in the submission.

*Where does this scope concern come from?* This may be based on the generic statement on board AI (in the first part)
> Current AI research mainly treats explanations as a means for model inspection. Yet, this neglects findings from human psychology that describe the benefit of self-explanations in an agent’s learning process….

I would suggest the author to rewrite these parts (also in the intro) by limiting the scope into the explainability ML only.

- **Concerns about "why this method works"**

I have read the revised paper, and it does not claim to provide a theoretically principled method. Instead, it proposes a practical approach, Self-Explaining (LSX), inspired by self-refining AI. In the introduction, the paper introduces the rationale behind self-refining with numerous references supporting psychology and AI. Therefore, I believe this paper implicitly illustrates why this method works, as it mimics human self-reflection behaviour, a concept explored in AI.

- **Concerns about "whether the method is scalable to most machine learning applications"**

In the experiments, the authors conducted three different LSX instantiations with image and text modalities. These experiments align with the claims made in the abstract and introduction. Specifically, the abstract and introduction claim experimental results in three settings with various datasets and evaluation metrics. Therefore, I think it does not need to fully justify the proposed method's scalability to the majority of machine learning applications, as it does not claim this in the paper.

- **Concerns about "more experimental evidence for the claims made and, if possible, some theoretical justification"**

As discussed earlier, the paper does not make a generic claim for all models but restricts its scope to several tasks. While adding more experiments and theories would be perfectly beneficial, I feel the current version's empirical evidence accurately supports the claim, satisfying the acceptance criteria.

### 5. Decision
Based on the discussions and the review history, I recommend acceptance with minor revisions. The followings are the suggestions for the minor revision
- **Terminology** Consider changing the term “framework” to “workflow” to more accurately reflect the nature of the proposed approach.
- **Scope Adjustment** It is advisable to narrow the scope to focus specifically on the explainability context rather than the broader AI context.
- **Discussion of Broader Limitations** Apart from addressing specific technical limitations, the authors are encouraged to discuss broader limitations, including: (1) The gap between human self-reflection and algorithmic implementations. (2) Theoretical justifications for the proposed approach. (3) Challenges in scaling the approach to a wider range of machine learning applications.

**Audience:**

Yes

**Claims And Evidence:**

Yes. But some minor points should be addressed, see the comments for details.

---

> ### Author Response · Authors · 2024-08-30
> **Revision containing final suggestions**
>
> Thank you so much for the great news! We particularly want to thank the reviewers for their valuable time and suggestions and the AE for coordinating the fruitful discussions and their high-quality and refreshingly thorough meta-review! We have updated the manuscript based on the final suggestions.
>
> However, we are uncertain about the exact final steps and, for now, have uploaded an anonymous version with the requested changes color-coded in blue. If these changes adhere to the requested modifications we would appreciate if the AE could let us know. We would then upload the final un-anonymous version.